# The *CDH1* c.1901C>T Variant: A Founder Variant in the Portuguese Population with Severe Impact in mRNA Splicing

**DOI:** 10.3390/cancers13174464

**Published:** 2021-09-04

**Authors:** Rita Barbosa-Matos, Rafaela Leal Silva, Luzia Garrido, Ana Cerqueira Aguiar, José Garcia-Pelaez, Ana André, Susana Seixas, Sónia Passos Sousa, Luísa Ferro, Lúcia Vilarinho, Irene Gullo, Vitor Devezas, Renata Oliveira, Susana Fernandes, Susy Cabral Costa, André Magalhães, Manuela Baptista, Fátima Carneiro, Hugo Pinheiro, Sérgio Castedo, Carla Oliveira

**Affiliations:** 1i3S, Instituto de Investigação e Inovação em Saúde, Rua Alfredo Allen 208, 4200-135 Porto, Portugal; amatos@ipatimup.pt (R.B.-M.); slealrafaela@gmail.com (R.L.S.); ana.c.aguiar@chedv.min-saude.pt (A.C.A.); jpelaez@ipatimup.pt (J.G.-P.); aandre@ipatimup.pt (A.A.); sseixas@ipatimup.pt (S.S.); ssousa@ipatimup.pt (S.P.S.); irene.gullo12@gmail.com (I.G.); fcarneiro@ipatimup.pt (F.C.); hpinheiro@ipatimup.pt (H.P.); scastedo@ipatimup.pt (S.C.); 2Ipatimup, Instituto de Patologia e Imunologia Molecular da Universidade do Porto, Rua Júlio Amaral de Carvalho 45, 4200-804 Porto, Portugal; 3Doctoral Programme on Cellular and Molecular Biotechnology Applied to Health Sciences (BiotechHealth), ICBAS-Institute of Biomedical Sciences Abel Salazar, University of Porto, 4050-313 Porto, Portugal; 4PhD Programme on Health Data Science, Faculty of Medicine, University of Porto, 4200-319 Porto, Portugal; 5CHUSJ, Centro Hospitar e Universitário de São João, Alameda Prof. Hernâni Monteiro, 4200-319 Porto, Portugal; luzia.garrido@chsj.min-saude.pt (L.G.); luisa.ferro@chsj.min-saude.pt (L.F.); lucia.vilarinho@chsj.min-saude.pt (L.V.); vitor.devezas7@gmail.com (V.D.); renata.oliveira@chsj.min-saude.pt (R.O.); susycosta@sapo.pt (S.C.C.); amag1976@gmail.com (A.M.); manuela.batista@chsj.min-saude.pt (M.B.); 6Department of Clinical Pathology, Centro Hospitalar Entre Douro e Vouga, Santa Maria da Feira, 4520-211 Santa Maria da Feira, Portugal; 7Doctoral Programme on Biomedicine, Faculty of Medicine, University of Porto, 4200-319 Porto, Portugal; 8Faculty of Medicine of the University of Porto, Alameda Prof. Hernâni Monteiro, 4200-319 Porto, Portugal; sf@med.up.pt; 9Department of Internal Medicine, Centro Hospitalar Tâmega e Sousa, 4564-007 Penafiel, Portugal

**Keywords:** HDGC, *CDH1*, founder effect, missense variants, cryptic splicing

## Abstract

**Simple Summary:**

An unexpectedly high number of early-onset diffuse gastric and lobular breast cancer in apparently unrelated families carrying the same *CDH1* c.1901C>T variant (formerly known as missense p.A634V) in Northern Portugal suggested a founder effect in this region. We demonstrated that c.1901C>T is a truncating variant triggered by cryptic splicing, calculated its mutational age, and characterized the tumour spectrum and age of onset in affected families.

**Abstract:**

Hereditary diffuse gastric cancer (HDGC) caused by *CDH1* variants predisposes to early-onset diffuse gastric (DGC) and lobular breast cancer (LBC). In Northern Portugal, the unusually high number of HDGC cases in unrelated families carrying the c.1901C>T variant (formerly known as p.A634V) suggested this as a *CDH1*-founder variant. We aimed to demonstrate that c.1901C>T is a bona fide truncating variant inducing cryptic splicing, to calculate the timing of a potential founder effect, and to characterize tumour spectrum and age of onset in carrying families. The impact in splicing was proven by using carriers’ RNA for PCR-cloning sequencing and allelic expression imbalance analysis with SNaPshot. Carriers and noncarriers were haplotyped for 12 polymorphic markers, and the decay of haplotype sharing (DHS) method was used to estimate the time to the most common ancestor of c.1901C>T. Clinical information from 58 carriers was collected and analysed. We validated the cryptic splice site within *CDH1*-exon 12, which was preferred over the canonical one in 100% of sequenced clones. Cryptic splicing induced an out-of-frame 37bp deletion in exon 12, premature truncation (*p.Ala634ProfsTer7*), and consequently RNA mediated decay. The haplotypes carrying the c.1901C>T variant were found to share a common ancestral estimated at 490 years (95% Confidence Interval 445–10,900). Among 58 carriers (27 males (M)–31 females (F); 13–83 years), DGC occurred in 11 (18.9%; 4M–7F; average age 33 ± 12) and LBC in 6 females (19.4%; average age 50 ± 8). Herein, we demonstrated that the c.1901C>T variant is a loss-of-function splice-site variant that underlies the first *CDH1*-founder effect in Portugal. Knowledge on this founder effect will drive genetic testing of this specific variant in HDGC families in this geographical region and allow intrafamilial penetrance analysis and better estimation of variant-associated tumour risks, disease age of onset, and spectrum.

## 1. Introduction

Evidence for familial predisposition to gastric cancer (GC) is present in approximately 10% of all GC cases, with 1–3% proven to be hereditary [1]. The first heritable syndrome recognized as linked to GC predisposition is known as hereditary diffuse gastric cancer (HDGC) [2]. This autosomal dominant tumour risk syndrome predisposes to diffuse gastric cancer (DGC) in both sexes and to lobular breast cancer (LBC) in females [3]. The definition of HDGC evolved along the years, as well as the clinical criteria driving genetic testing. Currently, *CDH1* germline variants, namely single nucleotide variants and large deletions, are its best characterized predisposing factor [2,4,5].

Clinical practice guidelines were defined to identify at-risk individuals for DGC and/or LBC, and were recently updated in 2020 (Appendix A) [5]. The present genetic screening criteria are less restrictive than the previous ones regarding age limits and consider personal or family history of cleft lip and/or cleft palate as evidence if co-occurring with DGC [3,5,6,7].

The disease risk associated with causal *CDH1* germline variants considerably varies, depending on the selected study, which exposes the limited understanding of disease penetrance in HDGC families [8]. Several factors can influence risk calculations, namely family ascertainment, the type of *CDH1* variants analysed, and the GC incidence in the home country of these families [5]. The bias in most reported studies has mainly reflected the inclusion of families that comply with clinical criteria and present high prevalence of DGC and/or LBC [8,9,10,11]. When considering both unascertained and HDGC families together, predicted disease risks decrease, as demonstrated by Roberts et al [5,8].

According to the ACMG/AMP criteria, only *CDH1* variants classified as pathogenic or likely pathogenic are considered actionable in the HDGC context [5,12]. For management of HDGC patients carrying actionable *CDH1* variants, prophylactic removal of the whole stomach to prevent DGC is recommended. The risk for LBC can be managed through bilateral risk-reducing mastectomy [5]. *CDH1* variant carriers may also undergo DGC surveillance by annual endoscopy with multiple biopsies and/or annual breast magnetic resonance imaging (MRI) if unfit or if refusing prophylactic measures [5].

Variants of unknown significance are a challenge for HDGC management, and *CDH1* missense variants are a particular hurdle. Despite being frequently detected in families with HDGC criteria, their actionability and ability to predispose to disease is hard to predict [12,13]. Some *CDH1* variants, initially considered to be missense, were later proven be protein-truncating, with clear functional consequences due to cryptic splicing activation, and therefore suggested as pathogenic or likely pathogenic [12,14]. This evidence reflects the importance of deepening the study of missense variants, particularly their potential to induce cryptic splicing and promote premature truncation. This will improve variant classification and have a clear impact in prediction of variant actionability.

In certain regions of Northern Portugal, more specifically in the district of Porto (an administrative division of the Portuguese territory currently subdivided in 18 counties—see map in Figure 1A, the economic growth, infrastructure development, and industrial expansion achieved in the last century have led to a recent populational increment and the settlement of large communities in this area, which established therein businesses and families. Several Porto counties, namely those located on the northeastern side, are former rural regions where local populations were originally reduced in size and geographically isolated. These features might have led to significant shifts in variant frequency (genetic drift). If local populations were originally reduced in size and isolated, these features might have led to significant shifts in variant frequency, predisposing to a phenomenon known as a founder effect. By definition, this phenomenon is associated with a reduction in genetic diversity where colonization occurs associated with a population decrease, migration, or isolation [15]. Therefore, a founder effect might explain the clustering, in the district of Porto, of several apparently unrelated families carrying the *CDH1* c.1901C>T variant (all living in neighbouring parishes located in the northeast of Porto in Northern Portugal). This explanation would be particularly compelling if the variant’s mutation age precedes the timing of the expansion of this population (the end of the 19th century to the beginning of the 21st century). However, if this hypothesis holds true, it is important to highlight that the recent population rise could have also promoted the spread and frequency increase of a previously rare and deleterious variant [16,17]. The high incidence of GC in Northern Portugal [18,19] and the privileged industrial and economic situation of this region, which allowed inhabitants to settle and succeed, further support a possible clustering of *CDH1* germline variant carriers predisposed to early-onset DGC and/or LBC in this region [14,20].

The c.1901C>T variant, located in *CDH1* exon 12 (Chr16: 68822190, GRCh38) was first described by Vécsey-Semjén et al. as a somatic out-of-frame mutation identified in a colon cancer cell line [21]. Previous in silico predictions and in vitro studies (cell lines and minigene assays) reported the c.1901C>T change to create cryptic-splicing and premature truncation [14,21] (Figure 1B). However, the outcome of the missense variant (p.A634V) overexpression in *CDH1*-negative cell lines is still widely used to support its deleteriousness [22]. To clarify this, we aimed at demonstrating, by using RNA obtained from HDGC germline carriers, that the *CDH1* c.1901C>T variant produces exclusively premature truncation codon (PTC)-bearing transcripts. This provides additional evidence to support the classification of the c.1901C>T variant as pathogenic (in ClinVar, currently classified as pathogenic/likely pathogenic—VCV000012244.5). This knowledge will improve clinical management provided to carriers of this variant and their families. Herein, by following families and individuals carrying the *CDH1* c.1901C>T variant, we characterized the first *CDH1*-related founder effect in Northern Portugal, traced its origin, and portraited tumour spectrum and age of onset in carrying families.

## 2. Materials and Methods

### 2.1. Study Design and Data Collection

The collection of clinical data and blood samples from patients and families was approved by the appropriate Ethics Committee from each of the centres participating in this work, i3S/IPATIMUP and Centro Hospital Universitário de São João (CHUSJ) in Porto, Portugal. Family histories and clinical data were obtained with informed consents and herein pseudoanonymized.

### 2.2. Biological Material (DNA and RNA) Collection from CDH1 c.1901C>T Variant Bearing Families

Genomic DNA was extracted from 200 µL whole blood using NZY Tissue gDNA Isolation Kit (NZYTech, Lisboa, Portugal) according to the manufacturer’s instructions. The fragment containing the *CDH1* c.1901C>T variant was amplified by standard PCR (NZYTaq DNA polymerase, NZYTech, Lisboa, Portugal) with the following primers: forward: 5′-CACCCGGTTCCATCTACCTTT and reverse: 5′-CCTTTCCAACCCCTCCCTACT at an annealing temperature of 60 °C. Direct sequencing of the PCR products was performed by direct Sanger sequencing (BigDye^®^ Terminator v3.1 Cycle Sequencing Kit, ThermoFisher, Waltham, MA, USA) using an automated sequencer (ABI PRISM^®^ 3130 Genetic Analyzer, Applied Biosystems).

RNA was isolated from peripheral blood mononuclear cells (PBMCs). PBMCs were isolated using Ficoll^®^ Paque Plus (Sigma-Aldrich, St. Louis, MO, USA) according to the manufacturer’s instructions. The pellet was resuspended in 300 μL of Lysis/Binding Buffer from the kit mirVana™ miRNA Isolation Kit (Life Technologies, Carlsbad, CA, USA). This kit was used for total RNA isolation according to the manufacturer’s instructions. RNA concentration was measured with Qubit™ RNA HS Assay Kit (Life Technologies, Carlsbad, CA, USA).

### 2.3. cDNA Synthesis and PCR Amplification

Reverse transcriptase (RT)-PCR was performed with 10 ng of input RNA using the SuperScript™ VILO™ cDNA Synthesis Kit (Life Technologies, Carlsbad, CA, USA). The reaction was performed using the following conditions: 5 × RT reaction mix, 10 × enzyme mix, 10 ng of RNA in a final volume of 10 μL, for 30 min at 42 °C and 5 min at 85 °C. The sequence between *CDH1* exon 11 and 13 was amplified with Qiagen^®^ Multiplex PCR Kit (Qiagen, Hilden, Germany) according to the manufacturer’s instructions and with the primers and conditions mentioned in Appendix A. Amplification was confirmed by electrophoresis in a 2% agarose gel, and PCR products were purified with the GE Kit according to “Protocol for purification of DNA from solution or an enzymatic reaction” and eluted in a final volume of 20 μL of DNase/RNase free water (Invitrogen, Carlsbad, CA, USA). The DNA concentration of the purified product was measured with NanoDrop 2000c Spectrophotometer (Thermo Scientific, Wilmington, NC, USA).

### 2.4. Cloning and Colony PCR

PCR products were cloned into pCR^®^ 2.1-TOPO^®^ vector, TOPO^®^ TA Cloning^®^ Kit, (Invitrogen, Carlsbad, CA, USA) according to the manufacturer’s instructions and using Mach1-T1 competent bacteria. The colony PCR with Universal M13 primers was performed in 80 colonies with Qiagen^®^ Multiplex PCR Kit (Qiagen, Hilden, Germany) according to the manufacturer’s instructions. M13 forward and reverse primers were provided by the TOPO^®^ TA Cloning^®^ Kit, (Invitrogene, Carlsbad, CA, USA) and used in the reaction at 0.4 μm (primers and PCR conditions are described in Appendix A). Amplification was confirmed by electrophoresis of the products in a 2% agarose gel, and the selected amplified products were sequenced using the primers and conditions described in Appendix A.

### 2.5. CDH1 Allelic Specific Expression (ASE) by SNaPshot

Confirmation of the heterozygous state at the single nucleotide polymorphism (SNP) rs1801552 was performed with polymerase chain reaction (PCR) sequencing in germline DNA. cDNA and gDNA from rs1801552 C/T F2 (family proband) were amplified with SNP flanking primers. All primers and PCR conditions are available in Appendix A. Purified PCR products were submitted to SNaPshot reaction using SNaPshot Multiplex Kit (Applied Biosystems, Foster City, CA, USA) and a single-base-extension (SBE)-specific primer following manufacturer’s instructions. Final products were purified with FastAP™ Thermosensitive Alkaline Phosphatase (Thermo Scientific, Wilmington, NC, USA) and run in the ABI PRISM^®^ 3130xl Genetic Analyzer (Applied Biosystems, Foster City, CA, USA). Peak Scanner™ software (ThermoFisher, Waltham, MA, USA) was used for peak analysis.

### 2.6. Histopathological Analysis

Four-micrometre sections were used for haematoxylin and eosin (HE) staining and immunohistochemistry (IHC). IHC for E-cadherin (clone 4A2C7, ThermoFisher, Waltham, MA, USA) was performed with Ventana BenchMark XT (Ventana Medical Systems, AZ, USA) automated immunostainer according to the manufacturer’s guidelines. Tissue sections were deparaffinized and rehydrated. After antigen retrieval, sections were incubated with a primary antibody against E-cadherin, and 3,3′-diaminobenzidine (DAB) was used as a chromogen. Finally, the slides were counterstained with haematoxylin, and coverslips were placed.

### 2.7. CDH1 c.1901C>T Variant Classification According to ACMG Guidelines

The actionability of *CDH1* c.1901C>T variant in the context of HDGC was evaluated using the ACMG/AMP Variant Curation Guidelines for the Analysis of Germline *CDH1* Sequence Variants [12]. These rules consider the variant’s impact at the RNA and protein level, relevant in vitro studies, frequency in families fulfilling HDGC criteria, the presence of *de novo* variants in LBC/DGC cases, and frequency in control populations. This allows classifying *CDH1* variants into one of five categories: pathogenic and likely pathogenic (actionable in HDGC), VUS (variant unknown significance), likely benign, and benign.

### 2.8. Identification of the Recombination Point in 16q22.1 Chromosome’s Region, Haplotyping of Different Polymorphic Markers

Thirty-eight DNA samples provided by CHUSJ were PCR amplified for the polymorphic markers (upstream and downstream of *CDH1*). Primers used and expected amplicons are described in Appendix A. For microsatellite analyses, 1 μL PCR products were mixed with 9.5 μL Hi-Di™ Formamide (Applied Biosystems, Foster City, CA, USA) and 0.5 μL of molecular size standard dye GeneScan™ 120 LIZ™ or GeneScan™ 500 ROX™ (Applied Biosystems, Foster City, CA, USA) depending on the molecular size of the amplified fragment. The mixture was run in BI PRISM^®^ 3130xl Genetic Analyzer (Applied Biosystems, Foster City, CA, USA) and analysed with Peak Scanner™ software (ThermoFisher Waltham, MA, USA). For SNP analysis, the PCR products were purified by FastAP™ thermosensitive alkaline phosphatase and exonuclease I (Thermo Scientific, Wilmington, NC, USA), following the manufacturer’s instructions, and used for Sanger sequencing with the BigDye^®^ Terminator Cycle v3.1 (Applied Biosystems, Foster City, CA, USA), according to the manufacturer’s instructions. The purified products were run in ABI PRISM^®^ 3130xl Genetic Analyzer (Applied Biosystems, Foster City, CA, USA), and the analysis was performed using Mutation Surveyor^®^ software (SoftGenetics, State College, PA, USA).

### 2.9. Characterization and Estimation of Age of the CDH1 c.1901C>T Variant

The time to most recent common ancestor (TMRCA) of *CDH1* c.1901C>T variant was estimated by the Decay of Haplotype Sharing MAPping (DHSMAP) software [23]. Four families carrying the c.1901C>T variant were characterized for 12 polymorphic markers (2 single nucleotide polymorphisms (SNPs) and 10 short tandem repeats (STRs)) located at different distances from *CDH1* and spanning a region of 19.2 Mb (16:62304172 to 16:81414947 GRCh38-hg38 assembly). All haplotypes were inferred according to their segregation within families (Appendix A). Since this work was based exclusively on family data, we limited identical copies of the same segregating haplotype to single chromosomes, as previously done [24]. Based on this condition, two different runs were performed with DHSMAP: a more conservative analysis (A) was first assumed, which allowed only the inclusion of single *CDH1* c.1901C>T chromosomes, one per each family (F1, F2, F3, and F7); while a second analysis (B) permitted the inclusion of two derived haplotypes resulting from mutation or recombination events at the D16S514 marker in families F7 (F7-A carrier) and F2 (F2-F carrier), respectively. Moreover, since for these individuals the D16S514 marker allele in phase with the *CDH1* c.1901C>T variant could not be accurately inferred from family segregation, two alternative haplotype configurations were considered in DHSMAP analysis. The genetic distance (23.2 cM for the 19.2 Mb region surrounding *CDH1*) used in DHSMAP for c.1901C>T calculations was extracted from chr.16 recombination maps generated for the Iberian sample of the 1000 Genomes project Phase I (http://www.internationalgenome.org/data#download; NCBI FTP Site: http://ftp-trace.ncbi.nih.gov/1000genomes/ftp/; http://ftp-trace.ncbi.nih.gov/1000genomes/ftp/technical/working/20130507_omni_recombination_rates/, accessed on 4 May 2018). Mutation rates were assumed as 0.001 for microsatellites [25] and 2.5 × 10^−8^ for SNPs [26]. In total, haplotypes from the 4 mutated chromosomes, 2 combinations of descendent affected haplotypes, and 18 control chromosomes were used for TMRCA estimation.

## 3. Results

### 3.1. The CDH1 c.1901C>T Variant Generates Cryptic Splicing within Exon 12, Leading to Premature Truncation and Decreased CDH1 RNA Levels

To evaluate the consequence of the *CDH1* c.1901C>T variant, we isolated RNA from PBMCs from patients F2 and F4-A, both carriers of the variant (methodological strategy depicted in Figure 2A). F2 was a female patient and the Index case of family F2 (partial pedigree—Appendix A). She was diagnosed with multiple (*n* = 5) LBC foci at 49 years old and submitted to prophylactic gastrectomy at 51 years old. Histopathological analysis revealed three foci of signet ring cell carcinoma of the stomach (SRCC) invading the lamina propria (pT1a) of the stomach. At 53 years old, she underwent contralateral risk reduction mastectomy, which revealed lobular intraepithelial neoplasia. Individual F4-A is the son of the Index in family F4 (partial pedigree—Appendix A). He was submitted to prophylactic total gastrectomy at 19 years old. Histopathological analysis of the gastrectomy specimen revealed 24 SRCC foci.

Colony PCR of cDNA transcripts using primers flanking exons 12 and 13, where both the *CDH1* c.1901C>T variant and rs1801552 SNP lie, showed PCR products with two different sizes in both patients (Figure 2B and Appendix A). While most colonies in both patients (86% and 85%) presented a product of the expected size (674 bp), herein called large product or LP, a smaller product was also observed in both patients (14% and 15%), herein called small product or SP (637 bp) (Figure 2B and Appendix A). As predicted, sequencing of the LP band revealed a canonical *CDH1* mRNA fragment encompassing a single allele carrying both the wild-type (WT) c.1901C and the frequent SNP rs1801552C. Sequencing of the SP band showed a deletion of 37 bp spanning from the c.1900 position (in exon 12) to the initial nucleotide of exon 13 (Appendix A). This result is compatible with the presence of a cryptic splice site at the position c.1900 of *CDH1* mRNA, lying immediately before the c.1901C>T variant position, that is preferred over the canonical splice site if the variant c.1901C>T is present. 

This cryptic splicing leads to the formation of an out-of-frame transcript that harbours a premature stop codon (PTC) 21 nucleotide after the c.1900 position. Additionally, all sequenced SP colonies displayed a T at SNP rs1801552 (position c.2076 of the *CDH1* coding sequence) (Figure 2C and Appendix A). This showed that rs1801552C occurred in the same allele as the WT c.1901C, and that rs1801552T occurred in the same allele as the mutant c.1901T. The colony PCR also showed a lower abundance of the mutant c.1901T allele in comparison with the WT allele, which may suggest downregulation of the PTC-bearing mutant allele by nonsense-mediated mRNA decay. To determine if the two alleles were differentially expressed in variant carriers, we applied a primer extension analysis quantitative method (SNaPshot) in both gDNA and cDNA from PBMCs of F2’s proband and used the heterozygous rs1801552 SNP as a marker. In gDNA, both C and T alleles were represented by equivalent peaks showing their heterozygosity. In contrast, in cDNA, the rs1801552C WT allele was consistently over-represented as compared to the rs1801552T mutant allele (Figure 2D). This transcript abundance analysis supported an active degradation of the PTC-bearing allele leading to its reduction by approximately 66%, which is consistent with the truncating nature of the *CDH1* c. 1901C>T variant (Figure 2D). The proposed mechanism is depicted in Figure 2E.

Further supporting the truncating nature of the *CDH1* c.1901C>T variant was the fact that E-cadherin protein expression was absent in both DGC and LBC tumour samples from families F1 and F2, as depicted in Figure 3.

After confirming that *CDH1* c.1901C>T is a bona fide splice-site variant, we used ACMG/AMP guidelines for its classification (12). A PVS1_strong criterium was attributed, as RNA analysis has been previously performed in a cancer cell line (18) supporting our current data. We also considered for classification clinical information from 13 families fulfilling HDGC criteria and carrying this variant (reported to us by different labs), as well as two families fulfilling HDGC criteria and reported in the literature (14, 27). This analysis returned a PS4_strong score. Additionally, this variant was absent in control populations (GnomAD) (PM2_moderate), and it was demonstrated to segregate with DGC and LBC in multiple family members of several families (PP1_strong). Altogether, these criteria supported the classification of the *CDH1* c.1901C>T variant (VCV000012244.5) as pathogenic.

### 3.2. Age Estimation of CDH1 c.1901C>T Variant through Haplotyping and Identification of the Recombination Point in 16q22.1 Chromosome

From 1999 to 2018, we found seven apparently unrelated families or isolated early-onset DGC cases bearing the *CDH1* c.1901C>T variant, and from 2018 to 2021, we identified two additional HDGC families carrying the same variant (details provided in Figure 1). We could confirm that all nine families were originated from the same region of Northern Portugal (Porto district), although they did not know each other. Given the low likelihood that a very rare variant would have occurred by chance in the same geographical region, we set out to demonstrate that this was a founder effect. To this end, we studied 12 polymorphic markers spanning a region of 19.2 Mbp in four HDGC families with available gDNA (F1, F2, F3, and F7), including affected carriers, asymptomatic carriers, and noncarriers (Figure 4A and Appendix A). Haplotype analysis and representative heredograms of two of these families (F1 and F2) are depicted with the results of the sequenced markers (Figure 4B,C). All remaining pedigrees with corresponding haplotypes are depicted in Appendix A. 

Four *CDH1* c.1901C>T chromosomes were identified (one per family) (Figure 4B). Families F1 and F2 were found to be identical across the entire 19.2 Mb/23.2 cM region studied. The other two chromosomes from F3 and F7 shared the same ancestral haplotype at shorter extensions; in F7, the affected chromosome carried the same marker alleles in a 6.8 Mb/5.1 cM region (D16S514 to D16S3067), and in F3, in a 19.1 Mb/18.4 cM region (D16S318 to D16S3098) region. Notably, in F7 and F2, two events of mutation or recombination were detected as modifying former haplotypes configurations at D16S514 marker (F7-A and F2-F) (Figure 4B,C and Appendix A).

Three other families (F4, F5, and F6) were also haplotyped, but only for 6/12 polymorphic markers because of a lack of material and a limited number of samples (Appendix A), which hampered the complete haplotype analysis. Nevertheless, besides having in common the c.1901C>T variant, these families shared several STR alleles among themselves and with the other four families widely haplotyped as described above.

### 3.3. CDH1 c.1901C>T a Founder Variant with Approximately 500 Years

We used two approaches to estimate the age of *CDH1* c.1901C>T variant (Figure 4B and Appendix A). In Analysis A, we performed a more conservative analysis, and a single affected chromosome per family was considered. In Analysis B, the previously identified chromosomes were used together with two derived haplotypes in DHSMAP. The TMRCA estimates converged in both analyses to 19.58 generations ago, or 489.64 years, given a generation of roughly 25 years. The results of both approaches diverged only in their 95% confidence intervals (CI), which were larger in the 4-chromosome analysis (Analysis A): 17.8–436 generations or 445–10,900 years. The inclusion of two derived haplotypes (Analysis B) had a 95% CI of 449.75–1492 years, if assuming a recombination event, and 449.75–1475.5 years if a mutational event has occurred instead. This means that the original carrier of this mutation, the founder, lived in the 16th century, and that the mutation has segregated in Northern Portuguese populations for nearly 500 years. At the moment, the distribution and frequency of this variant in the population is unpredictable.

### 3.4. Clinical Presentations in HDGC Families Carrying CDH1 c.1901C>T

Assuming, from the data described above, that all families from Northern Portugal carrying the very rare *CDH1* c.1901C>T truncating variant had a common ancestor, we next present the clinical data from all carriers together.

To date, 134 individuals from nine families underwent carrier testing for the *CDH1* c.1901C>T variant, and 43.3% (58/134) were proven carriers of this variant. Of the 58 carriers, 31 were females (53.4%). We depict the numbers of clinically diagnosed DGC cases in Table 1 and LBC cases in Table 2, as well as related numbers of carriers submitted to risk reduction surgeries and, of these, those revealing DGC and/or LBC foci during surveillance.

DGC was diagnosed in 11/58 (18.9%) variant carriers, four males (M) with an average age of onset of 27 ± 7 years old and seven females (F) with an average age of onset of 37 ± 13 years old. Five out of the 11 DGC patients (2M/3F) were diagnosed in the context of disease surveillance (endoscopic biopsy) (Table 1).

Twenty-six carriers (10M/16F) decided to undergo risk reduction gastrectomy, and in 21/26 (80.8%; 8M/13F), DGC foci were found in the gastrectomy specimen.

LBC, which only occurs in females (31/58 carriers), was diagnosed in 6/31 (19.4%) female carriers at an average age of onset of 50 ± 8 years old (Table 2). Two out of the six LBC patients were diagnosed during disease surveillance in the context of the high-risk consultation. From the female carriers that performed prophylactic mastectomies, 4/8 (50%) presented LBC foci (Table 2). Importantly, two of these women were already previously diagnosed with LBC on the contralateral breast; consequently, only unilateral prophylactic mastectomy was performed. A third woman, also previously diagnosed with LBC, underwent tumourectomy before proceeding to bilateral mastectomy. Therefore, only one of the four female carriers had no previous lesions when bilateral risk reducing mastectomy was performed. Three female carriers were found with both DGC and LBC foci in risk reduction surgical specimens (Table 1).

We further analysed the average age and age range of carriers upon disease onset, genetic test, or last surveillance, and separated the cohort (*n* = 58 carriers) in four groups: (1) carriers presenting clinical expression of DGC and/or LBC; (2) asymptomatic carriers with subclinical disease (foci found in RRG or RRM); (3) carriers without foci detected in RRG and RRM specimens; and (4) asymptomatic carriers not submitted to risk reduction surgery. The average age of onset of the first group, including all carriers with DGC and/or LBC (*n* = 16), was 39 ± 13.98 years old (age range: 18–61 years old). The average age at which subclinical disease was identified in the second group of asymptomatic carriers (*n* = 18), upon risk-reduction surgery, was 36 ± 14 years old (age range: 14–63). In the third group, carriers without subclinical disease (*n* = 4), the average age at risk reduction surgery was 40 ± 14.9 years old (age range: 23–59). The remaining carriers were also asymptomatic and decided to avoid RRG and/or RRM (*n* = 20). In this last group, follow-up was carried out for DGC in both genders and for LBC in females, starting at an average age of 44 ± 22.7 years old (age range: 18–83), and to date none have developed clinical disease. These data support a high variability in age of disease onset and incomplete penetrance in these families. c.1901C>T is, therefore, a deleterious but low-penetrant variant, probably because of co-segregation with either protective or disease-predisposing genetic modifiers.

## 4. Discussion

In this study, we used biological material from HDGC patients to formally prove that the single nucleotide variant in *CDH1* (c.1901C>T) they carry in their germline, earlier identified as the missense variant A634V (Figure 1), is indeed a bona fide splice-site variant (r.1900_1936del; p. Ala634ProfsTer7). This finding supports its classification as pathogenic according to ACMG/AMP guidelines for *CDH1* variant classification [12]. Like several other *CDH1* variants [12,14], the c.1901C>T induces cryptic splicing within *CDH1* exon 12, frameshifting, and premature truncation (Figure 2 and Appendix A). This is believed to target the variant-bearing allele for degradation, most likely through the mRNA nonsense-mediated decay (NMD) pathway. NMD is a quality-control translational step that selectively degrades mRNAs harbouring premature termination (nonsense) codons [27,28]. Therefore, normal E-cadherin protein levels are sustained exclusively by the wild-type allele in normal tissues, until a second somatic hit occurs to abolish *CDH1* function in HDGC-target organs. As compared to transcripts generated by genuine missense variants, which produce E-cadherin molecules and retain residual activity, the complete loss of E-cadherin immunoexpression in DGC and LBC from carriers, accompanying the strong downregulation (at least 66% in relation to the wild-type allele) of PTC-bearing transcripts produced by the mutant allele, may explain the aggressive phenotypes seen in patients carrying the c.1901C>T germline variant [13,29]. We published on similar levels of downregulation of PTC-carrying alleles, using RNA extracted either from peripheral blood lymphocytes (PBLs) or normal gastric mucosa, from patients carrying germline nonsense variants in *CDH1* from different families in two of our previous papers [28,30]. The lack of complete downregulation of the PTC-bearing allele by NMD may result from a delay in degradation after transcription. Indeed, as none of the nonsense variants studied resulted in transcription inactivation, the transcription level of the mutant allele is likely similar to that of the wild-type. After transcription, splicing and the pioneer round of translation initiate, as well as degradation by NMD of PTC-carrying alleles [31]. This delay likely explains the residual detection of c.1901T variant-bearing transcripts in PBL samples from the c.1901C>T variant carriers. Regarding the impact that complete or incomplete downregulation of the mutant allele may have for disease phenotype and severity, very little is known. However, it has been documented that certain genetic modifiers, and interindividual variability in NMD efficiency between patients carrying identical mutations, may lead to differences in the disease severity and clinical phenotype [32].

Although a previous study reported splicing impairment related to this variant in a colorectal cancer cell line [21] and using minigene analysis [14], the current report was the first to assess the impact of c.1901C>T in germline RNA from HDGC patients, supporting its reclassification as a truncating variant. This information should replace the fundaments for the variant’s pathogenicity, currently available at the ClinVar entry VCV000012244.5, which mainly describe findings on the functional impact of a p.A634V missense-bearing protein.

Recently, Corso et al. performed an extensive literature review on *CDH1* missense variants described to occur in HDGC patients in order to assess their relevance and implications for clinical management of this disease [22]. This study, albeit one of great relevance, considered the c.1901C>T variant as a missense variant, and the functional consequence of the p.A634V mutant protein. This formally contaminated data analysis from pure missense variants with data referring to bona fide truncating variants.

Another important subject is the choice of functional studies to understand the impact of *CDH1* variants. Indeed, we and others have predicted a deleterious effect for the *CDH1* p.A643V missense variant, supposedly generated by the c.1901C>T variant. These predictions were based on overexpression of a cDNA construct with a single nucleotide variant, mimicking the c.1901C>T variant, in *CDH1*-negative cell lines [14,21,33]. However, given the data herein reported, such a missense-bearing transcript and mutant E-cadherin protein do not occur in nature. This brings into discussion the absolute need to assess the impact, in silico and at the RNA level, of variants classified as missense, prior to the design of the most appropriate set of functional analyses. This will be fundamental for variant interpretation and will certainly improve variant classification and variant carriers’ management. The molecular reclassification of some missense variants may indeed change our knowledge on DGC and LBC frequencies by type of variant and, consequently, on disease penetrance estimation in HDGC.

Several publications have now demonstrated that a considerable fraction of variants initially assigned as missense are indeed splice-site variants. One such publication studied *NF1*, a gene causing neurofibromatosis type 1 [34]. After applying in silico predictive tools to 41 variants previously classified as missense variants, the authors demonstrated that over 45% of them were shown to affect splicing and claimed that such variants have been frequently underscored if not analysed in depth. These observations were further supported by studies using next-generation sequencing technology [32] and high-throughput splicing reporter assays [35] to unravel the effect on splicing of single nucleotide variants, including missense, in the von Willebrand factor gene and *POU1F1*, respectively. Altogether, there is a need for improvement of in silico tools to predict cryptic splicing, and for the development of guidelines for formal analysis of splicing defects induced by SNVs, particularly synonymous and missense variants. These will certainly improve the current knowledge of the molecular mechanisms causing disease and will facilitate variant interpretation.

The *CDH1* c.1901C>T variant was previously described in unrelated HDGC families in Portugal, England, and New Zealand [14,36,37,38]. Its unusually high frequency among early-onset DGC and LBC cancer cases, and families fulfilling HDGC criteria, in a specific region of Northern Portugal (Porto district) [18,19], led us to explore the occurrence of a potential founder effect.

Haplotyping data from four families carrying the c.1901 C>T variant showed carriers to share at least 1.8 Mbp of chromosome 16. Strikingly, families F1 and F2 shared the same haplotype in all the polymorphic markers studied (total of 19.2 Mbp, Figure 4B). These results indicate that F1 and F2 carriers inherited and maintained the same genetic information from a past common ancestor. Carriers from family F7 differed from those from families F1 and F2 in five STR markers, showing that F7 family diverged from the most likely common ancestor present in F1 and F2 by a recombination event close to D16S3095 (Figure 4B). This was demonstrated first by a 6 bp difference in D16S3095 (154 bp in F7 vs. 148 bp in F1 and F2), corresponding to three mutational steps (three dinucleotide repeats) [39], and second because downstream markers also deviated from the common ancestral haplotype. Family F3 distinguished itself from the F1 and F2 in a single 5′-external polymorphic marker (Figure 4B). Because of the absence of upstream markers, we could not infer whether the affected chromosome detected in F3 resulted from mutation or recombination. In our study, we identified two derived chromosomes (F7-A carrier, and F2-F carrier) that differed within their families for the marker D16S514. This polymorphic variability may have resulted either from an acquired mutation, particularly if it only differed from the ancestral allele by one mutational step (single repeat—116 bp allele), or from recombination (120 bp in F7 and 124 bp in F2). Unfortunately, we could not discriminate which was the case here, as the derived haplotype could not be accurately inferred.

The two DHSMAP analysis (A and B) converged to the same time frame, 490 years ago, only diverging in their confidence intervals (CI). When the two descendent haplotypes, which differed only by a mutation or recombination event at D16S514, were added, the CI was narrower. However, as explained above, it was not possible to infer these chromosomes properly. Using a more conservative hypothesis, our CI was very broad, indicating future improvements to age estimation by including a higher number of samples from different c.1901C>T variant-carrying HDGC families.

This study estimated that the c.1901C>T variant has been segregating within the Portuguese population since the 16th century, close to the year 1528. This date was calculated by subtracting the estimated *CDH1* allele age (490 years) from the year 2018, when this analysis was performed (1528 = 2018−490 years) and assuming a 25-year generation time. These data may also indicate that more families are likely to be at risk and remain unidentified to date.

There is a limitation in this study regarding the number of carrier individuals used for the RNA analysis; however, as carriers herein analysed were from different families, we believe our data were still robust. Additionally, for the purpose of inferring haplotypes, the lack of samples from complete trios could interfere with the presented estimations. For this reason, families F4, F5, and F6 were excluded from the TMRCA analysis (Appendix A).

One of the critical challenges in HDGC clinical management is to predict whether a carrier will develop the disease or not. Although proven pathogenic, this variant seems to present a low penetrance in HDGC families, as depicted by the DGC and LBC frequencies among carriers. From the 58 mutation carriers identified, only 27.6% (16/58) were clinically affected, and only one female was diagnosed with both LBC and DGC (Table 1 and Table 2). Moreover, predicting whether premalignant lesions could lead to carcinomas or remain indolent is a challenge to solve in the future. Indeed, signet-ring cell carcinoma (SRCC) foci can remain indolent without proliferation, never leading to clinical disease, but some foci may also evolve, becoming very aggressive and often killing the patient. It has been reported in other studies that the number of cancer foci found in prophylactic gastrectomy specimens ranged from 1 to 487, and the size from < 0.1 mm to 16.0 mm, without a clear relation to patients’ age or gender [40,41]. Therefore, some of the carriers herein presented may develop SRCC foci and/or clinically expressed cancer later in their lives. Additionally, unrecognized genetic modifiers, either protective or predisposing to disease development, may be conditioning the apparently low disease penetrance in the setting herein described.

## 5. Conclusions

The *CDH1* c.1901C>T variant, widely studied for its functional impact as the missense variant p.A634V, has been demonstrated to be a bona fide truncating variant (r.1900_1936del; p.Ala634ProfsTer7), supporting its classification as pathogenic according to ACMG/AMP *CDH1* guidelines.

The *CDH1* c.1901C>T variant was herein proven as the first *CDH1*-associated founder effect in Portugal, still restricted to a specific region of the northeastern part of the Porto district and aged approximately 490 years.

In our cohort of nine HDGC families carrying the *CDH1* c.1901C>T variant, the frequency of DGC and LBC was 18.9% and 19.4%, respectively. However, the prophylactic measures applied resulted in the identification of carcinoma foci in almost 81% of the carriers who underwent prophylactic gastrectomy and 50% of those submitted for prophylactic mastectomies, demonstrating the efficiency of these measures.

Despite the clearly incomplete penetrance of disease in these families, the c.1901C>T variant’s early age of onset and aggressiveness highlights the importance and supports the use of intensive surveillance and prophylaxis in asymptomatic carriers of this variant.

The herein-established clinical actionability of the c.1901C>T variant, and the demonstration of a founder effect in this geographical region, empowered the monitoring of at-risk individuals and diagnosis of DGC and LBC during surveillance.

Publishing these data will increase awareness among clinicians on the need to adjust HDGC testing criteria in high-risk geographical areas such as Northern Portugal.

## Figures and Tables

**Figure 1 cancers-13-04464-f001:**
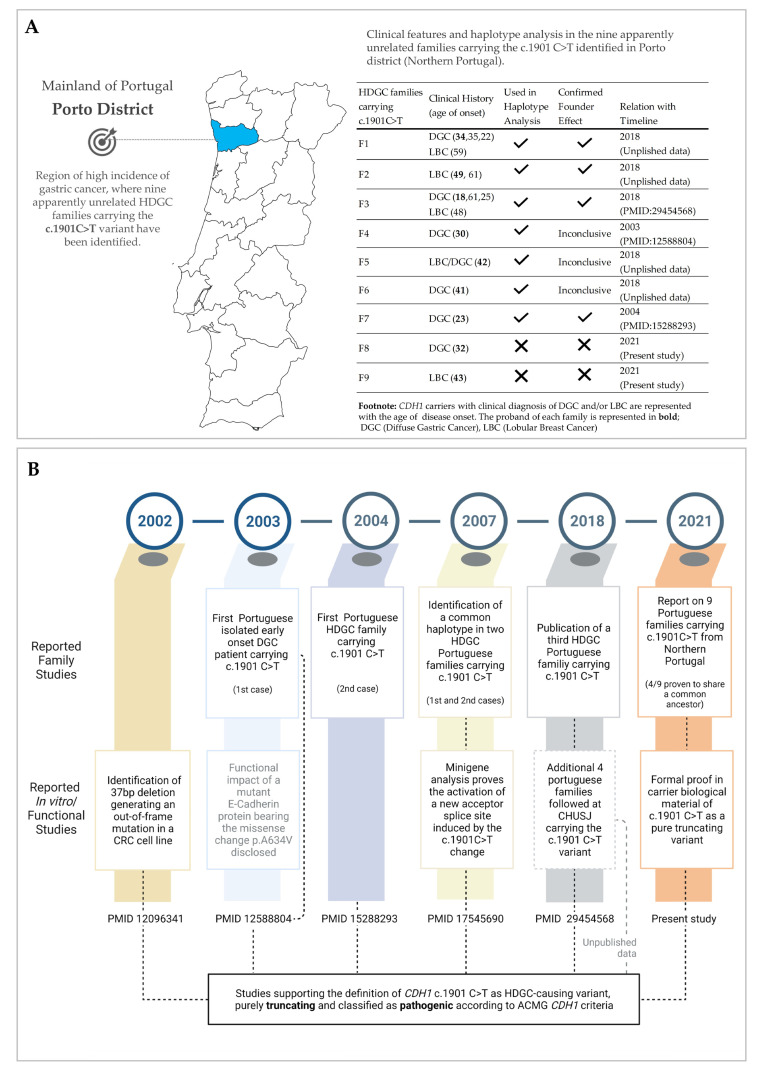
Timeline of the discovery of a *CDH1*-related founder effect in Northern Portugal: the *CDH1* c.1901C>T variant. (**A**) Map of the mainland of Portugal highlighting the Porto district in Northern Portugal, a region of high incidence of GC where the nine c.1901C>T variant carriers’ families were identified; and clinical features of carrier families (right hand side table). (**B**) Timeline of discoveries related to the c.1901C>T *CDH1* variant: from pathogenicity to carrier families. CRC, colorectal carcinoma; DGC, diffuse gastric cancer; HDGC, hereditary diffuse gastric cancer; CHUSJ, Centro Hospitalar Universitário de S. João; ACMG/AMP, American College of Medical Genetics and Genomics and the Association for Molecular Pathology *CDH1* criteria [12].

**Figure 2 cancers-13-04464-f002:**
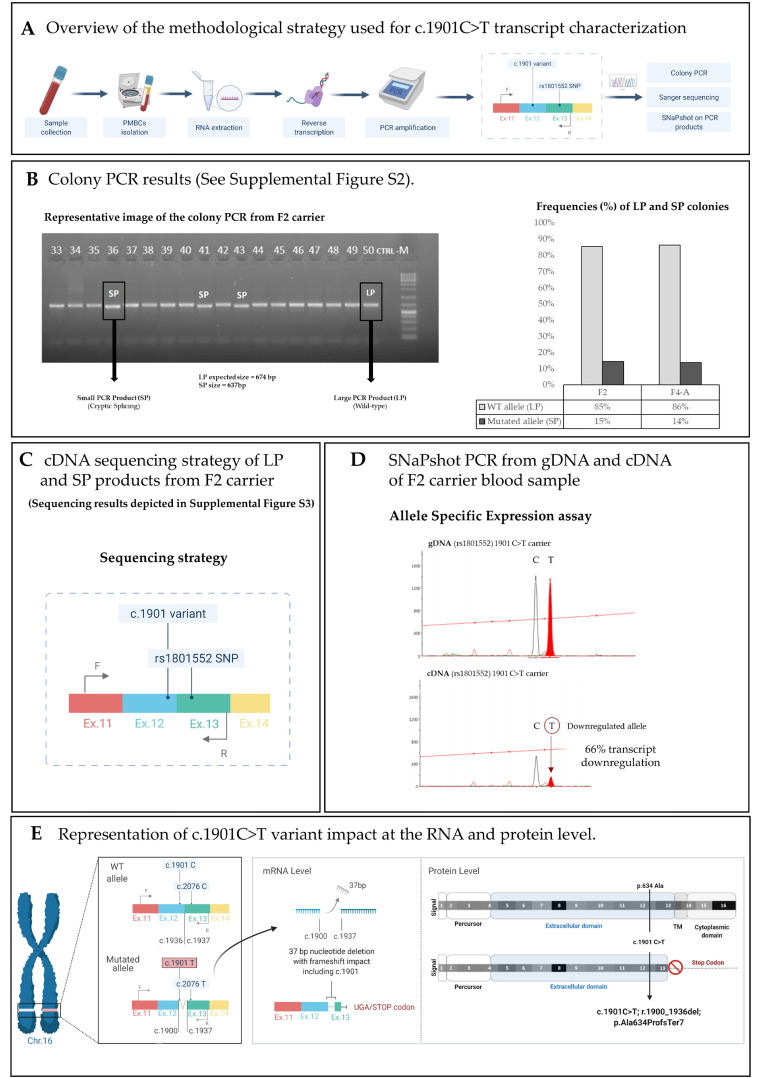
c.1901C>T *CDH1* variant generates a cryptic splice site and a transcript with premature truncation. (**A**) Overview of the methodological strategy used for c.1901C>T transcript characterization; (**B**) Electrophoresis of the colony PCR after cloning cDNA of 2 carriers; (**C**) cDNA sequencing strategy of LP and SP products from F2 carrier; (**D**) SNaPshot PCR from gDNA and cDNA of F2 carrier blood sample; (**E**) representation of c.1901C>T impact at the RNA and protein level. PMBCs, peripheral blood mononuclear cells; gDNA, genomic DNA; cDNA, complementary DNA; WT, wild type.

**Figure 3 cancers-13-04464-f003:**
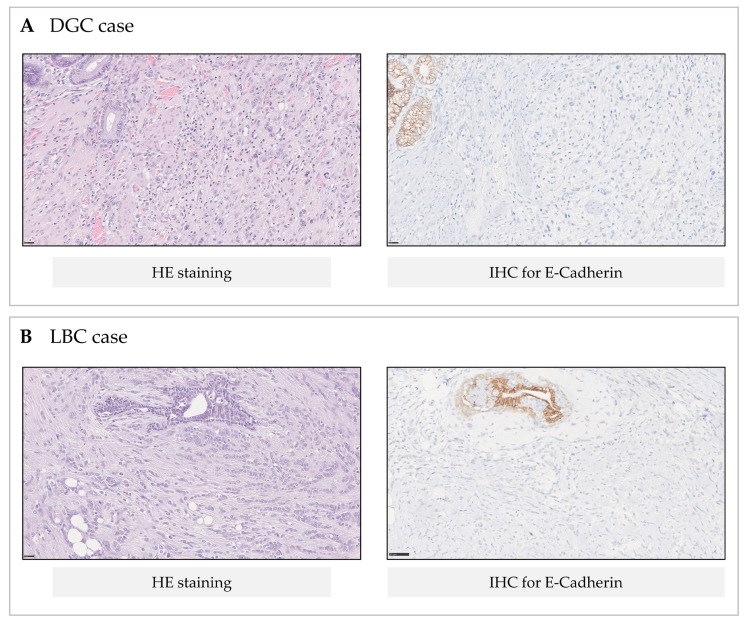
HE (haematoxylin and eosin) and IHC (immunohistochemistry) staining for DGC (F1 carrier) and LBC (F2 carrier) tumours from *CDH1* c.1901C>T variant carriers. (**A**) Poorly cohesive (diffuse) gastric cancer composed by pleomorphic and spindle cells infiltrating the submucosa. Note the presence of residual gastric glands (top right of the image). E-cadherin IHC shows loss of E-cadherin expression in tumour cells and preserved membranous expression in adjacent non-neoplastic epithelium. (**B**) Classic lobular breast cancer composed by uniform cells lacking cohesion, dispersed, and arranged in single-file linear cords. E-cadherin IHC shows loss of E-cadherin expression in tumour cells, while normal membranous expression is retained in non-neoplastic ducts (upper part of the image).

**Figure 4 cancers-13-04464-f004:**
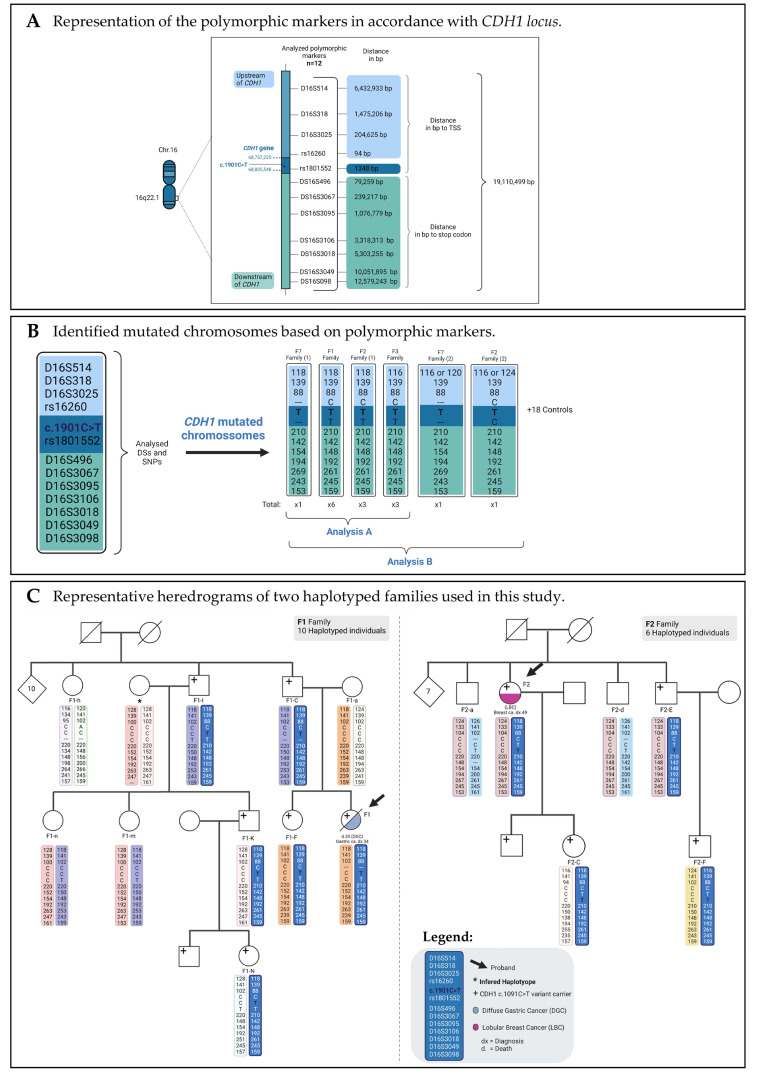
Polymorphic markers used and haplotype results for characterizing c.1901C>T mutated families: (**A**) Representation of the studied polymorphic markers in accordance with *CDH1* locus; (**B**) Identified *CDH1* mutated chromosomes used in the two different runs performed with DHSMAP for time to most recent common ancestor (TMRCA) estimation; (**C**) Selected heredograms of two representative families used in this study. TSS, transcription start site; bp, base pairs; STRs, short tandem repeats.

**Table 1 cancers-13-04464-t001:** Overview of stomach-related clinical presentations in *CDH1* c.1901C>T variant carriers from nine HDGC families.

	Total nº of Tested Individuals	Total nº of Carriers	DGC Cases	Mean Age of DGC Diagnosis (±SD)	RRG	RRG with Lesions
Males	60	27	4	27 ± 7	10	8
Females	74	31	7 *	37 ± 13	16	13 ^#^
Total	134	58	11	33 ± 12	26	21

Footnote: * One out of seven females presented DGC and LBC; ^#^ from 13 females bearing lesions in RRG, 3 also presented LBC. DGC, diffuse gastric cancer; RRG, risk reduction gastrectomy; LBC, lobular breast cancer; Age of diagnosis is represented by the mean of the affected carrier’s age ± SD.

**Table 2 cancers-13-04464-t002:** Overview of breast-related clinical presentations in *CDH1* c.1901C>T variant carriers from nine HDGC families.

Number of Females Tested	Total nº of Carriers	LBC Cases	Mean Age of LBC Diagnosis (± SD)	RRM	RRM with Lesions	Lesions in both RRG and RRM
74	31	6	50 ± 8	8	4 *	3 ^#^

Footnote: * Two out of four female patients with RRM lesions were submitted to unilateral mastectomy, since they were previously diagnosed with LBC in the other breast; ^#^ all three patients were previously diagnosed with LBC and presented carcinoma foci in the RRM (on the other breast) and in the RRG. LBC, lobular breast cancer; RRM, risk reduction mastectomy; Age of diagnosis is represented by the mean of the affected carrier’s age ± SD.

## Data Availability

Clinical and genetic data from patients herein described are not publicly available due to ethical, legal and privacy issues.

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
