# Peer review of "The CDH1 c.1901C>T Variant: A Founder Variant in the Portuguese Population with Severe Impact in mRNA Splicing"

_cancers, 2021, doi:10.3390/cancers13174464_

Round 1
Reviewer 1 Report
Nice paper.
1) I think it is worth noting that commercial testing labs do consider this at least a VLP (per ClinVar) as this leads the reader to believe that the variant is a solid VUS.
2) Language regarding mastectomy should be changed as it should be consistent with Blair paper (mastectomy can be considered). The language in the introduction reads recommended.
3) Paper would be enhanced by IHC on the tissues from specimens to see if this was lost in the tumors. This would solidify functional data. Not sure if this was done.
4) The font in Figure 3 (especially 3C) is very small and hard to read even when zoomed in. Please make it a little larger if possible.
5) The breast data is interesting, but table 1 is a bit challenging to follow with both gastrectomy and mastectomy data. Perhaps splitting it if possible would flow better.
6) Please define "foci". Is there a size?
7) Capitalize P (Portuguese) in Figure 1
Author Response
Cover Letter and Point-by-point reply to Reviewers
Porto, August 26th, 2021
Manuscript ID: cancers-1341518
Revised Title: “The CDH1 c.1901C>T variant: a founder variant in the Portuguese population with severe impact in mRNA splicing”
Dear Editor-in-Chief of CANCERS, Prof. Dr. Samuel C. Mok,
Dear Editors of the special issue of Cancers (ISSN 2072-6694): "Hereditary Gastric Cancer—Molecular Basis and Diagnosis"
We greatly appreciate this comprehensive peer review of our manuscript that allowed us to improve the clarity and quality of our paper.
We herein submit an updated version of the manuscript taking into consideration all the comments from Reviewers 1 and 2. We submit a clean version with each alteration highlighted as a comment on the side of the text, numbered according to the Reviewers questions, which were listed and answered in a complementary file with point-by-point responses to all comments (Letter_Reviewers_26Aug_2021)
All the new tables and figure legends (cancers-1341518-MainText-Revised-CleanVersion_CO) were revised according to the MDPI Journal style, for references implemented. Moreover, we verified all the requirements for the figures.
We hope that the Editor-in-Chief of Cancers, Reviewers and the Editorial Manager find these amendments to our manuscript adequate and sufficient to proceed with the publication process in this special issue.
Sincerely,
Carla Oliveira
Reply to Reviewer 1 comments and suggestions:
- 1) I think it is worth noting that commercial testing labs do consider this at least a VLP (per ClinVar) as this leads the reader to believe that the variant is a solid VUS.
Response 1: We acknowledge and agree with Reviewer 1 observation, therefore we changed the manuscript text to clarify this point. The CDH1 c.1901C>T variant, although considered pathogenic or likely pathogenic in all ClinVar submissions, is still considered and reported as a missense variant in several publications (e.g.: Corso et al, 2021; Gullo I. et al 2018; Melo S. et al 2017). In particular, most functional analysis performed to date on this variant use the premise that the variant c.1901C>T produces a mutant CDH1 protein bearing the p.A634V alteration and test its effects in non-human overexpressing cell lines. For classification purposes using ACMG guidelines, this may bring confusion for those classifying the variant, as it is likely that these functional analyses are considered. Herein, we wish to demonstrate, by using patient-derived biological material, that the c.1901C>T variant does not produce peptides bearing the aminoacid change p.A634V, and therefore should not be considered a missense variant. This is a bona fide truncating variant (r.1900_1936del; p.Ala634ProfsTer7), and its classification is Pathogenic, according to the ACMG/AMP CDH1 guidelines.
We have modified part of the introduction and conclusion to address the Reviewer's point:
Page 4, Rev1 Comment 1: “The c.1901C>T variant, located in CDH1 exon 12 (Chr16: 68822190, GRCh38) was first described by Vécsey-Semjén et al. as a somatic out-of-frame mutation identified in a colon cancer cell line [21]. Previous in silico predictions and in vitro studies (cell lines and minigene assays) reported the c.1901C>T change to create cryptic-splicing and premature truncation [14,21]. However, many publications on the functional consequences of this variant still use the outcome of the missense variant (p.A634V) overexpression in CDH1-negative cell lines to support its deleteriousness [22]. To formally change this view, we aimed at demonstrating, by using RNA obtained from HDGC germline carriers, that the CDH1 c.1901C>T variant produces exclusively premature truncation codon (PTC)-bearing transcripts. This provides additional evidence to support the current classification of the c.1901C>T variant in ClinVar (VCV000012244.5 - pathogenic/Likely pathogenic).”
Page 18, Rev 1- Comment 1: “The CDH1 c.1901C>T variant, widely studied for its functional impact as the missense variant p.A634V, has herein been proved to be a bona fide truncating variant (r.1900_1936del; p.Ala634ProfsTer7), supporting its classification as Pathogenic according to ACMG/AMP CDH1 guidelines.”
- 2) Language regarding mastectomy should be changed as it should be consistent with Blair paper (mastectomy can be considered). The language in the introduction reads recommended.
Response 2: We thank the Reviewer 1 observation, and we agree with it. We re-wrote a short sentence of the prophylactic measures offered to HDGC patients to improve the clarity of the message.
We have modified part of the introduction to address the Reviewer's point:
Page 3, Rev 1- Comment 2: “Management of HDGC patients carrying actionable CDH1 variants recommends prophylactic removal of the whole stomach to prevent DGC. The risk for LBC can be managed through bilateral risk-reducing mastectomy [5]. CDH1 variant carriers may also undergo DGC surveillance by annual endoscopy with multiple biopsies and/or annual breast magnetic resonance imaging (MRI), if unfit or if refusing prophylactic measures [5].
- 3) Paper would be enhanced by IHC on the tissues from specimens to see if this was lost in the tumors. This would solidify functional data. Not sure if this was done.
Response 3: We acknowledge and agree with Reviewer 1 observation. Therefore, we altered Figure 4 (Page 14), including both HE and IHC results for E-Cadherin staining from a DGC and an LBC patient carrying the c.1901C>T variant, to replace former Figure 4. Accordingly, we added to the Methodology a subsection entitled “2.6 IHC staining” and we present the obtained results with a sentence in the “Results” section.
Page 7 (Material and Methods) Rev 1 Comment 3: “2.6. Histopathological Analysis: Four-micrometer sections were used for haematoxylin and eosin (H&E) staining and immunohistochemistry (IHC). IHC for E-cadherin (clone 4A2C7) was performed with Ventana BenchMark XT automated immunostainer according to the manufacturer’s guidelines. Tissue sections were de-paraffinized and rehydrated. After antigen retrieval, sections were incubated with a primary antibody against E-cadherin, and 3,3′-diaminobenzidine (DAB) was used as a chromogen. Finally, the slides were counterstained with hematoxylin, and coverslips were placed”.
Page 9 (results) Rev 1 - Comment 3: “Further supporting the truncating nature of the CDH1 c.1901C>T variants is the fact that E-cadherin protein expression is absent in both DGC and LBC tumour samples from families F1 and F2, respectively, as depicted in Figure 4.”
Figure 4. HE and IHC staining for DGC and LBC tumours from CDH1 c.1901C>T variant carriers. A) Poorly cohesive (diffuse) gastric cancer composed by pleomorphic and spindle cells infiltrating the submucosa. Note the presence of residual gastric glands (top right of the image). E-cadherin IHC shows loss of E-cadherin expression in tumour cells and preserved membranous expression in adjacent non-neoplastic epithelium. B) Classic lobular breast cancer composed by uniform cells lacking cohesion, dispersed, and arranged in single-file linear cords. E-cadherin IHC shows loss of E-cadherin expression in tumour cells, while normal membranous expression is retained in non-neoplastic ducts (upper part of the image).
- 4) The font in Figure 3 (especially 3C) is very small and hard to read even when zoomed in. Please make it a little larger if possible.
Response 4: We acknowledge and agree with Reviewer 1 observation. We increased the font size in Figure 3 (Page 12).
- 5) The breast data is interesting, but table 1 is a bit challenging to follow with both gastrectomy and mastectomy data. Perhaps splitting it if possible would flow better.
Response 5: We thank the Reviewer 1 observation. This table was splitted in two parts, one with the DGC and gastrectomy data and a second one with the LBCs and mastectomy data. These new tables are now called Table 1 and Table 2, and are depicted below. Tables 1 and 2 were also added in the revised version of the manuscript (both in page 15).
Table 1. Overview of stomach-related clinical presentations in CDH1 c.1901C>T variant carriers from nine HDGC families.
Footnote: *1/7 females presenting DGC and LBC; # From 13 females bearing lesions in RRG, 3 also presented LBC. DGC, Diffuse Gastric Cancer; RRG, Risk Reduction Gastrectomy; LBC, Lobular Breast Cancer; Age of diagnosis is represented by the mean of the affected carrier’s age ± SD.
Table 2. Overview of breast-related clinical presentations in CDH1 c.1901C>T variant carriers from nine HDGC families.
Footnote: *2/4 female patients with RRM lesions performed unilateral mastectomy since they were previously diagnosed with LBC in the other breast; # All three patients were previously diagnosed with LBC and presented carcinoma foci in the RRM (in the other breast) and in the RRG. LBC, Lobular Breast Cancer; RRM, Risk Reduction Mastectomy; Age of diagnosis is represented by the mean of the affected carrier’s age ± SD.
- 6)Please define "foci". Is there a size?
Response 6: We understand the Reviewer 1 comment. The term “foci” refers to groups of cells which stand out from the surrounding tissue at the microscopic level and should be carefully interpreted as an indicative of a cancer precursor lesion. Signet-ring cell carcinoma (SRCC) foci can remain indolent without proliferation, never leading to clinical disease. On the other hand, some SRCC foci may also evolve, becoming very aggressive and often killing the patient. The mechanisms behind the transition of SRCC from indolent to an aggressive state are still vastly unknown. It has been previously reported that the number of cancer foci found in prophylactic gastrectomy specimens ranged from 1 to 487 and the size from <0.1 mm to 16 mm, without a clear relation to patients age or gender (PMID: 30014492; PMID: 18825658).
We added part of this information at the end of the discussion to clarify similar doubts from the readership and to address the Reviewer's point:
Page 18, Rev 1 - Comment 6: “Indeed, signet-ring cell carcinoma (SRCC) foci can remain indolent without proliferation, never leading to clinical disease, but some foci may also evolve, becoming very aggressive and often killing the patient. It has been reported in other studies that the number of cancer foci found in prophylactic gastrectomy specimens ranged from 1 to 487, and the size from <0.1 mm to 16.0 mm, without a clear relation to patients age or gender [40,41]. So, some of the carriers herein presented may develop SRCC foci and/or clinically expressed cancer later in their lives. Additionally, unrecognized genetic modifiers, either protective or predisposing to the disease development, may be conditioning the apparently low dis-ease penetrance in the setting herein described. “
- 7) Capitalize P (Portuguese) in Figure 1
Response 7: We thank the Reviewer 1 observation, and we performed the correction in Figure 1 (Page 5).
Figure 1. Timeline of the discovery of a CDH1-related founder effect in Northern Portu-gal: the CDH1 c.1901C>T variant. A) Map of the mainland of Portugal highlighting the Porto District in Northern Portugal, a region of high incidence of GC, where the nine c.1901C>T variant carriers’ families have been identified; and clinical features of carrier families (right hand side table). B) Timeline of discoveries related to the c.1901C>T CDH1 variant: from pathogenicity to carrier families; CRC, Colorectal Carcinoma; DGC, Diffuse Gastric Cancer; HDGC, Hereditary Diffuse Gastric Cancer; CHUSJ, Centro Hospitalar Universitário S. João; ACMG/AMP, American College of Medical Genetics and Genomics and the Association for Molecular Pathology CDH1 criteria [12].

Reviewer 2 Report
The manuscript from Barbosa-Matos et al, is a well written paper documenting the in vivo impact on splicing of a founder variant in the Northern Portugal. Although the variant they describe was already predicted in-silico and in vitro to alter splicing the in vivo confirmation they describe is a useful add in diagnostic setting. The authors should better define the population history of Northern Portugal because in its present form the manuscript is not really clear on the population genetics events responsible for the diffusion of this variant in the area.
Finally, approximately 40% of the references have been coauthored by one or more authors of this manuscript. Therefore, an expanded discussion including similar mechanisms in other genes/disease and supporting references will help in reducing this bias. It will also help in drawing conclusions that can be generalized and of interest to a higher number of journal reader.
Q1. The definition of founder effect is "The founder effect is the reduction in genetic variation that results when a small subset of a large population is used to establish a new colony". According to this definition the paragraph at page 2 lines 88-102 is a bit confusing. First authors refer to Northern Portugal: could they better specify the geographic boundaries of Northern Portugal? Second, in lines 95-97 the combination of founder effects with population rise would lead to an increase of the frequency of founder variants only if the recently populated region had few residents. In its actual form the sentence is not clear on what authors do mean. Could authors explain more in detail the history of population in Northern Portugal ?
Q2. Authors write that in recent years northern Portugal had a fast population rise. A founder effect should be seen if this wave of immigration came in large part from a very isolated community. Do they have evidence for this? In case add reference.
Q3. The authors write (page 2 line 97) of high GC incidence in Northern Portugal. Do they have supporting data for this? In case please cite. Finally the conclusion mention an unspecified clinical finding. What is that?
Figure 1. Timeline of studies on c.1901C>T mutation. What is the advancement between year 2003 and year 2004? Are the families they are referring to, different or is the same family with a different classification?
Page 5 line 217. please correct maker with marker
Page 6 second paragraph (lines 238-266).
Q4. Authors mention the SNP rs1801552 which is cosegregating with the mutation. Have the authors verified is the T allele of this SNP is altering splicing modulation sequences such as ESE, ESS, with available web algorithm?
Q5. While PTC bearing mutant allele would be subject to NMD still they do see some mutant RNA production and without using NMD-rescuing reagents (i.e. puromycin or PAXgene tubes). So the effect is partial. Further, authors should detail the ratio of down-regulation of the mutant RNA (approximately down to 30% when compared to wt in figure 2C).
Q6. page 8 line 287. As above please define northern Portugal (eventually add map).
DISCUSSION SECTION
In the discussion authors could expand the discussion on NMD effects possibly explaining why it appears to be partial in the data they show (see ref PMID 33928629 and PMID 33313762).
In the second paragraph of page 12 (line 396-401) authors may expand the discussion on the classification of missense variants affecting splicing (DOI 10.3390/genes9040216, PMID 30361419, PMID: 34270938)
Page 12 line 414.
In here I'm not sure that the 6 bp differences in D16S3095 results correspond to three different mutational steps. Misalignment and slippage errors could also consist of two dinucleotide repeats or three dinucleotide repeats, therefore the mutation steps are not necessarily three. If authors have reference on this topic please add.
Page 12 line 433
From where the exact year (1528) comes?
Although the mutation appears to be associated to a clinically aggressive phenotype it also show some incomplete penetrance (overall 17 DGC+LBC in 58 carriers). The authors acknowledge this in the last discussion paragraph. Do they have data on the median age of healthy carriers that might disclose some hints on a combination of lower penetrance and age dependent penetrance (in case even comparing with other pathogenic CDH1 variants)?
Author Response
Cover Letter and Point-by-point reply to Reviewers
Porto, August 26th, 2021
Manuscript ID: cancers-1341518
Revised Title: “The CDH1 c.1901C>T variant: a founder variant in the Portuguese population with severe impact in mRNA splicing”
Dear Editor-in-Chief of CANCERS, Prof. Dr. Samuel C. Mok,
Dear Editors of the special issue of Cancers (ISSN 2072-6694): "Hereditary Gastric Cancer—Molecular Basis and Diagnosis"
We greatly appreciate this comprehensive peer review of our manuscript that allowed us to improve the clarity and quality of our paper.
We herein submit an updated version of the manuscript taking into consideration all the comments from Reviewers 1 and 2. We submit a clean version with each alteration highlighted as a comment on the side of the text, numbered according to the Reviewers questions, which were listed and answered in a complementary file with point-by-point responses to all comments (Letter_Reviewers_26Aug_2021)
All the new tables and figure legends (cancers-1341518-MainText-Revised-CleanVersion_CO) were revised according to the MDPI Journal style, for references implemented. Moreover, we verified all the requirements for the figures.
We hope that the Editor-in-Chief of Cancers, Reviewers and the Editorial Manager find these amendments to our manuscript adequate and sufficient to proceed with the publication process in this special issue.
Sincerely,
Carla Oliveira
Reply to Reviewer 2 comments and suggestions:
1) The definition of founder effect is "The founder effect is the reduction in genetic variation that results when a small subset of a large population is used to establish a new colony". According to this definition the paragraph at page 2 lines 88-102 is a bit confusing. First authors refer to Northern Portugal: could they better specify the geographic boundaries of Northern Portugal?
Second, in lines 95-97 the combination of founder effects with population rise would lead to an increase of the frequency of founder variants only if the recently populated region had few residents. In its actual form the sentence is not clear on what authors do mean. Could authors explain more in detail the history of population in Northern Portugal ?
Response 1: We thank Reviewer 2 for these questions. In order to specify the geographic area related to the founder effect herein reported, we added in Figure 1 a panel depicting the map of Mainland Portugal and highlighting the district where a high number of c.1901C>T carriers have been identified (Figure 1, Panel A, represented in blue). The exact city is not mentioned to preserve patients' identity, confidentiality and due to Ethical restrictions related to patient’s data protection. To overcome this lack of specific geographical localization, we now provide a better description of the geographic region where families carrying the CDH1 c.1901C>T variant have been living for many years. All nine families have been found to live and work in a narrow area of Northern Portugal, within the district of Porto (Northeastern part of Porto) that, as requested, is now represented in a map (Figure 1).
As indicated in the text, the district of Porto comprises former rural and low mobility populations that in recent years (XX century) suffered great increments in the number of their inhabitants (up to 6-fold increases). Such demographic shifts from small size and isolated groups to expanded populations, moving around the extended geographic area of the Porto district, might have impacted the dispersal of a rare mutation like the CDH1 c.1901C>T. This event is currently better contextualized with the hypothesis of an earlier founder effect (lines 99-104).
In this respect, the age estimates achieved for the c.1901C>T variant point to an origin (or introduction) of this variant around 1528 A.D, four centuries before the rising of the Porto district population. This remark is now included in the discussion.
Page 17, Rev 2 - Comment 1: “This study estimates that the c.1901C>T variant has been segregating within the Portuguese population since the 16th century, and close to the year of 1528. This date was calculated by subtracting the estimated CDH1 allele age (490 years) to the year of 2018, when this analysis was performed (1528 = 2018 - 490 years), and assuming a 25 years generation time. These data may also indicate that more families are likely to be at risk and remain unidentified, to date.”
We also re-wrote the paragraph in the introduction and added two new references that refer specifically to the description of high incidence of gastric cancer in Northern Portugal, namely:
Castro, C., Antunes, L., Lunet, N., et al. 25, 472-480 (2016).Cancer incidence predictions in the North of Portugal: keeping population-based cancer registration up to date. European Journal of Cancer Prevention PMID: 26317384 DOI: 10.1097/CEJ.0000000000000199
Castro C., Peleteiro B, et al. "Trends in Gastric and Esophageal Cancer Incidence in Northern Portugal (1994-2009) by Subsite and Histology, and Predictions for 2015". https://doi.org/10.5301/tj.5000542. First Published August 23, 2016 PMID: 27647232 DOI: 10.5301/tj.5000542.
Page 3-4, Rev 2 - Comment 1: “In certain regions of Northern Portugal, more specifically in the district of Porto (an administrative division of the Portuguese territory currently subdivided in 18 counties – see map), the economic growth, infrastructures’ development, and industrial expansion achieved in the last century, led to a recent populational increment and settlement of large communities in this area, which established therein businesses and families. Several of Porto counties, namely those located in the northeastern side, were former rural regions where local populations were originally reduced in size and geographically isolated. These features might have led to significant shifts in variant frequency (genetic drift). If local populations were originally reduced in size and isolated, these features might have led to significant shifts in variant frequency, predisposing to a phenomenon known as a founder effect. By definition, this phenomenon is associated with a reduction in the genetic diversity where the colonization occurs associated with a population decrease, migration or isolation [15]. Therefore, a founder effect might explain the clustering, in the district of Porto, of several apparently unrelated families carrying the CDH1 c.1901C>T variant (all living in neighboring parishes located in the northeast of Porto in Northern Portugal), particularly if its mutation age precedes the timing of the expansion of this population (end of XIX to beginnings of XXI). However, if this hypothesis holds true, it is important to highlight that the recent population rise could have also promoted the spread and frequency increase of a previously rare and deleterious variant [16,17]. The high incidence of GC in Northern Portugal [18,19], the privileged industrial and economic situation of this region, that allowed inhabitants to settle and succeed, further supports a possible clustering of CDH1 germline variant carriers predisposed to early-onset DGC and/or LBC in this region [14,20].”
2) Authors write that in recent years northern Portugal had a fast population rise. A founder effect should be seen if this wave of immigration came in large part from a very isolated community. Do they have evidence for this? In case add reference.
Response 2: We thank Reviewer 2 for this question. We have partially addressed this point when replying to the previous comment. The answer to this particular point is rather complex. Indeed, the fast rise of the populations from the Porto district during the XX century is well documented by the official census of Portuguese population (“Instituto National de Estatística”). However, data for earlier periods, before the XIX, are not easily accessible. The historical information collected from several local organizations and information from the catholic church, point to a scattering of populations in small villages with the numbers of inhabitants fluctuating between one to four hundred.
We hope the reviewer understands these limitations and the impossibility of reporting better supported historical information due to data protection rules.
3) The authors write (page 2 line 97) of high GC incidence in Northern Portugal. Do they have supporting data for this? In case please cite. Finally, the conclusion mentions an unspecified clinical finding. What is that?
Response 3: We thank Reviewer 2 for the observation, and we addressed it already in the reply to comment 1 (please see above). To support the high incidence of gastric cancer in northern Portugal, and particularly in the Porto district, we added two references to the manuscript:
Added references:
Castro, C., Antunes, L., Lunet, N., et al. 25, 472-480 (2016).Cancer incidence predictions in the North of Portugal: keeping population-based cancer registration up to date. European Journal of Cancer Prevention PMID: 26317384 DOI: 10.1097/CEJ.0000000000000199
Castro C., Peleteiro B, et al. "Trends in Gastric and Esophageal Cancer Incidence in Northern Portugal (1994-2009) by Subsite and Histology, and Predictions for 2015". https://doi.org/10.5301/tj.5000542. First Published August 23, 2016 PMID: 27647232 DOI: 10.5301/tj.5000542
4) Question to Figure 1. Timeline of studies on c.1901C>T mutation. What is the advancement between 2003 and 2004? Are the families they are referring to, different or is the same family with a different classification?
Response 4: The publication of 2003, reports an isolated case of DGC at 32 years old, lacking family history of HDGC or other cancer-associated syndrome, carrying the CDH1 c.1901C>T variant. In 2004, a new publication reported a family fulfilling HDGC clinical criteria and a heavy family history of DGC carrying the exact same variant. These were the first two apparently unrelated individuals/families identified to carry the c.1901C>T variant. This has now been clarified in Figure 1.
We also took the opportunity to perform a few other amendments to improve the message of Figure 1 (Page 5).
5) Question to Page 5 line 217. please correct maker with marker
Response 5: We thank the Reviewer 2 for the observation, the correction was done on Page 8 line 250 of the new version of the manuscript (please see Rev 2- Comment 5).
6) Page 6 second paragraph (lines 238-266). Authors mention the SNP rs1801552 which is co-segregating with the mutation. Have the authors verified if the T allele of this SNP is altering splicing modulation sequences such as ESE, ESS, with available web algorithm?
Response 6: We thank the Reviewer 2 observation. The rs1801552 is a very frequent SNP in the all world populations and has been classified as a benign variant according to Clinvar and was revised by an FDA expert panel (https://www.ncbi.nlm.nih.gov/clinvar/variation/142770/). From a molecular standpoint, this variant has never been reported to impact protein function, mRNA expression or splicing and its frequency is equivalent in diseased and non-diseased populations, which is consistent with its benign classification. From our colony PCR sequencing, the vicinity of the rs1801552 SNP remained wild-type in all transcripts sequence, independently on whether the transcript was wild-type or mutant for the c.1901 position, as depicted in Supplemental Figure 3. Furthermore, we performed the same in silico analysis for the rs1801552 SNP, as used for the c.1901C>T alteration, using Netgene2 Server (http://www.cbs.dtu.dk/services/NetGene2/) (PMID: 2067018 DOI: 10.1016/0022-2836(91)90380-o) and we did not see a difference in the splicing prediction of both rs1801552 alleles.
7) While PTC bearing mutant allele would be subject to NMD still they do see some mutant RNA production and without using NMD-rescuing reagents (i.e. puromycin or PAXgene tubes). So the effect is partial. Further, authors should detail the ratio of down-regulation of the mutant RNA (approximately down to 30% when compared to wt in figure 2C).
Response 7: We thank and agree with Reviewer 2. Therefore, we considered this data, when describing the Allele-specific imbalance analysis results from Figure 2 Painel C (Page 10) and in the results section.
Page 9, Rev 2 - Comment 7: “This transcript abundance analysis supported an active degradation of the PTC bearing allele, leading to its reduction in approximately 66%, consistent with the truncating nature of the CDH1 c. 1901C>T variant.”
8) page 8 line 287. As above please define northern Portugal (eventually add map).
Response 8: As asked in previous comments, we added in Figure 1 (Page 5), a panel with a map of Portugal depicting the Porto district, known to present higher incidence of GC than the rest of the country, and also the region where the nine apparently unrelated families carrying the c.1901C>T reside.
DISCUSSION SECTION
9) In the discussion authors could expand the discussion on NMD effects possibly explaining why it appears to be partial in the data they show (see ref PMID 33928629 and PMID 33313762).
Response 9: Although a PTC is generated in the transcript derived from the c.1901T allele, the mRNA expression of this allele is 66% lower than the expression of the c.1901C wild-type allele (Figure 2 Painel X). Assuming the high efficiency of NMD in degrading PTC-bearing alleles, this may not seem an exceptional downregulation. However, we have seen similar results when using RNA extracted either from peripheral blood lymphocytes (PBLs) or normal gastric mucosa from patients carrying germline nonsense variants in CDH1 from different families, in two of our previous papers (PMID: 19965908; PMID: 18427545). The reason underlying the lack of complete downregulation of the PTC-bearing allele may result from a decalage between transcription and degradation. As none of these variants results in transcription inactivation, the expression of the mutant allele is likely similar to that of the wild-type. After transcription, splicing and the pioneer round of translation initiate, as well as degradation by NMD. This delay likely explains the residual detection of c.1901T variant-bearing transcripts.
To clarify this issue, we have added the following paragraph to the discussion:
Page 16, Rev 2- Comment 9: “We have published similar levels of downregulation of PTC-carrying alleles, when using RNA extracted either from peripheral blood lymphocytes (PBLs) or normal gastric mucosa, from patients carrying germline nonsense variants in CDH1 from different families, in two of our previous papers [28,30]. The reason underlying the lack of complete downregulation of the PTC-bearing allele by NMD, may result from a delay in degradation after transcription. Indeed, as none of the nonsense variants studied results in transcription inactivation, the transcription level of the mutant allele is likely similar to that of the wild-type. After transcription, splicing and the pioneer round of translation initiate, as well as degradation by NMD of PTC-carrying alleles [31]. This delay likely explains the residual detection of c.1901T variant-bearing transcripts in PBL samples from the c.1901C>T variant carriers. Regarding the impact that complete or incomplete downregulation of the mutant allele may have for disease phenotype and severity, very little is known, however it has been documented that certain genetic modifiers, and inter-individual variability in NMD efficiency between patients carrying identical mutations, may lead to differences in the disease severity and clinical phenotype [32].”
10) In the second paragraph of page 12 (line 396-401) authors may expand the discussion on the classification of missense variants affecting splicing (DOI 10.3390/genes9040216, PMID 30361419, PMID: 34270938)
Response 10: We understand the point of Reviewer 2 and we added a paragraph in the discussion alluding to the need of a wider characterization of the effects of apparent missense variants, and we cited the suggested references.
Pages 16 and 17, Revisor 2 - Comment 10: “Several publications have now demonstrated that a considerable fraction of variants, initially assigned as missense, are indeed splice-site variants. An example has been published for NF1, a gene causing Neurofibromatosis type 1 [34]. After applying in silico predictive tools to 41 variants previously classified as missense variants, authors demonstrated that over 45% of them were shown to affect splicing, and claim that such type of variants is frequently underscored if not analyzed in depth. These observations were further supported by studies using next generation sequencing technology [32] and high-throughput splicing reporter assays [35] to unravel the effect of single nucleotide variants, including missense, on splicing in other genes such as the von Willebrand factor gene and POU1F1, respectively. Altogether, there’s a need for improvement of in silico tools to predict cryptic splicing, and for the development of guidelines for formal analysis of splicing defects induced by SNVs, particularly synonymous and missense variants. These will certainly improve the current knowledge of the molecular mechanisms causing disease and will facilitate variant interpretation.”
11) Page 12 line 414. In here I'm not sure that the 6 bp differences in D16S3095 results correspond to three different mutational steps. Misalignment and slippage errors could also consist of two dinucleotide repeats or three dinucleotide repeats, therefore the mutation steps are not necessarily three. If authors have references on this topic please add.
Response 11: We understand and thank Reviewer 2 for this comment, and we provide a clarification below.
Although large mutational jumps might occur in short tandem repeats, the most common mutational process is through the gain or loss of one repeat fitting the postulated stepwise mutation model (PMID: 8401493; PMID: 8349120; PMID: 8159720).
In the specific case of family F7, and as indicated in lines 517-519, we assumed a recombination event as the source of haplotype divergence given that D16S3095, as well as the remaining downstream loci (D16S3106, D16S3018, D16S3049, D16S3098), all differ from the haplotypes identified in the other studied families. This assumption is therefore independent of the mutational model best suited to the D16S3095 polymorphic marker.
Still, as requested by the reviewer, we included in the text (lines 519-523) a reference (PMID: 8349120) for the assumed dinucleotide repeat mutational model.
Page 17, Rev 2 - Comment 11: “This can be demonstrated, firstly, by a 6 bp difference in D16S3095 (154 bp in F7 vs 148 bp in F1 and F2) corresponding to three mutational steps (three dinucleotide repeats) [39] and, secondly, because downstream markers deviate also from the common ancestral haplotype.”
12) Page 12 line 433: From where the exact year (1528) comes?
Response 12: We thank Reviewer 2 for the observation. The provided DHSMAP results (489,64 years) were obtained in 2018. Thus, subtracting 490 to 2018, we concluded that the TMRCA is from 1528. To make it clearer for the readers we detailed it in the text.
Page 17, Rev 2 - Comment 12: “This study estimates that the c.1901C>T variant has been segregating within the Portuguese population since the 16th century, and close to the year of 1528. This date was calculated by subtracting the estimated CDH1 allele age (490 years) to the year of 2018, when this analysis was performed (1528 = 2018 - 490 years), and assuming a 25 years generation time. These data may also indicate that more families are likely to be at risk and remain unidentified, to date.”
13) Although the mutation appears to be associated to a clinically aggressive phenotype it also shows some incomplete penetrance (overall 17 DGC+LBC in 58 carriers). The authors acknowledge this in the last discussion paragraph. Do they have data on the median age of healthy carriers that might disclose some hints on a combination of lower penetrance and age dependent penetrance (in case even comparing with other pathogenic CDH1 variants)?
Response 13: We thank Reviewer 2 for this observation and we altered the manuscript to address this subject. We compared the average age of and age-range (disease onset, genetic test, or last surveillance) for four groups in our cohort: 1) carriers presenting either DGC and/or LBC; 2) carriers with subclinical disease (foci in RRG or RRM); 3) carriers without foci detected in RRG and RRM specimens; 4) Asymptomatic carriers not submitted to risk reduction surgery.
Patients with either clinical or subclinical expression of disease presented on average age below 40 years old. A non-neglectable fraction of carriers (34,5%) rejected prophylactic measures and remain healthy to date with an average age of 42 years old, being the oldest non-affected carrier 83 years old. This supports the idea that c.1901C>T is a low penetrant variant, possibly regulated by other genetic modifiers that can either protect or promote disease development.
We added in the text the following paragraph:
Page 15, Rev 2 - Comment 13: “We further analyzed the average age and age-range of carriers upon disease onset, genetic test, or last surveillance, and separated the cohort (n=58 carriers) in four groups: 1) carriers presenting clinical expression of DGC and/or LBC; 2) carriers with subclinical disease (foci found in RRG or RRM); 3) carriers without foci detected in RRG and RRM specimens; 4) Asymptomatic carriers not submitted to risk reduction surgery. The average age of onset of the first group including all carriers with DGC and/or LBC (n=16) was 39 ± 13.98 years old [age range: 18-61 years old]. The average age at which subclinical disease was identified in the second group of asymptomatic carriers (n=18), upon risk-reduction surgery, was 36 ± 14 years old [age range: 14-63]. In the third group, carriers without subclinical disease (n=4), the average age at risk reduction surgery was 40 ± 14.9 years old [age range: 23-59]. The remaining asymptomatic carriers decided to avoid RRG and/or RRM (n=20). In these, follow-up was carried out either for DGC in both genders and for LBC in females, starting at a
Cover Letter and Point-by-point reply to Reviewers
Porto, August 26th, 2021
Manuscript ID: cancers-1341518
Revised Title: “The CDH1 c.1901C>T variant: a founder variant in the Portuguese population with severe impact in mRNA splicing”
Dear Editor-in-Chief of CANCERS, Prof. Dr. Samuel C. Mok,
Dear Editors of the special issue of Cancers (ISSN 2072-6694): "Hereditary Gastric Cancer—Molecular Basis and Diagnosis"
We greatly appreciate this comprehensive peer review of our manuscript that allowed us to improve the clarity and quality of our paper.
We herein submit an updated version of the manuscript taking into consideration all the comments from Reviewers 1 and 2. We submit a clean version with each alteration highlighted as a comment on the side of the text, numbered according to the Reviewers questions, which were listed and answered in a complementary file with point-by-point responses to all comments (Letter_Reviewers_26Aug_2021)
All the new tables and figure legends (cancers-1341518-MainText-Revised-CleanVersion_CO) were revised according to the MDPI Journal style, for references implemented. Moreover, we verified all the requirements for the figures.
We hope that the Editor-in-Chief of Cancers, Reviewers and the Editorial Manager find these amendments to our manuscript adequate and sufficient to proceed with the publication process in this special issue.
Sincerely,
Carla Oliveira
Reply to Reviewer 2 comments and suggestions:
1) The definition of founder effect is "The founder effect is the reduction in genetic variation that results when a small subset of a large population is used to establish a new colony". According to this definition the paragraph at page 2 lines 88-102 is a bit confusing. First authors refer to Northern Portugal: could they better specify the geographic boundaries of Northern Portugal?
Second, in lines 95-97 the combination of founder effects with population rise would lead to an increase of the frequency of founder variants only if the recently populated region had few residents. In its actual form the sentence is not clear on what authors do mean. Could authors explain more in detail the history of population in Northern Portugal ?
Response 1: We thank Reviewer 2 for these questions. In order to specify the geographic area related to the founder effect herein reported, we added in Figure 1 a panel depicting the map of Mainland Portugal and highlighting the district where a high number of c.1901C>T carriers have been identified (Figure 1, Panel A, represented in blue). The exact city is not mentioned to preserve patients' identity, confidentiality and due to Ethical restrictions related to patient’s data protection. To overcome this lack of specific geographical localization, we now provide a better description of the geographic region where families carrying the CDH1 c.1901C>T variant have been living for many years. All nine families have been found to live and work in a narrow area of Northern Portugal, within the district of Porto (Northeastern part of Porto) that, as requested, is now represented in a map (Figure 1).
As indicated in the text, the district of Porto comprises former rural and low mobility populations that in recent years (XX century) suffered great increments in the number of their inhabitants (up to 6-fold increases). Such demographic shifts from small size and isolated groups to expanded populations, moving around the extended geographic area of the Porto district, might have impacted the dispersal of a rare mutation like the CDH1 c.1901C>T. This event is currently better contextualized with the hypothesis of an earlier founder effect (lines 99-104).
In this respect, the age estimates achieved for the c.1901C>T variant point to an origin (or introduction) of this variant around 1528 A.D, four centuries before the rising of the Porto district population. This remark is now included in the discussion.
Page 17, Rev 2 - Comment 1: “This study estimates that the c.1901C>T variant has been segregating within the Portuguese population since the 16th century, and close to the year of 1528. This date was calculated by subtracting the estimated CDH1 allele age (490 years) to the year of 2018, when this analysis was performed (1528 = 2018 - 490 years), and assuming a 25 years generation time. These data may also indicate that more families are likely to be at risk and remain unidentified, to date.”
We also re-wrote the paragraph in the introduction and added two new references that refer specifically to the description of high incidence of gastric cancer in Northern Portugal, namely:
Castro, C., Antunes, L., Lunet, N., et al. 25, 472-480 (2016).Cancer incidence predictions in the North of Portugal: keeping population-based cancer registration up to date. European Journal of Cancer Prevention PMID: 26317384 DOI: 10.1097/CEJ.0000000000000199
Castro C., Peleteiro B, et al. "Trends in Gastric and Esophageal Cancer Incidence in Northern Portugal (1994-2009) by Subsite and Histology, and Predictions for 2015". https://doi.org/10.5301/tj.5000542. First Published August 23, 2016 PMID: 27647232 DOI: 10.5301/tj.5000542.
Page 3-4, Rev 2 - Comment 1: “In certain regions of Northern Portugal, more specifically in the district of Porto (an administrative division of the Portuguese territory currently subdivided in 18 counties – see map), the economic growth, infrastructures’ development, and industrial expansion achieved in the last century, led to a recent populational increment and settlement of large communities in this area, which established therein businesses and families. Several of Porto counties, namely those located in the northeastern side, were former rural regions where local populations were originally reduced in size and geographically isolated. These features might have led to significant shifts in variant frequency (genetic drift). If local populations were originally reduced in size and isolated, these features might have led to significant shifts in variant frequency, predisposing to a phenomenon known as a founder effect. By definition, this phenomenon is associated with a reduction in the genetic diversity where the colonization occurs associated with a population decrease, migration or isolation [15]. Therefore, a founder effect might explain the clustering, in the district of Porto, of several apparently unrelated families carrying the CDH1 c.1901C>T variant (all living in neighboring parishes located in the northeast of Porto in Northern Portugal), particularly if its mutation age precedes the timing of the expansion of this population (end of XIX to beginnings of XXI). However, if this hypothesis holds true, it is important to highlight that the recent population rise could have also promoted the spread and frequency increase of a previously rare and deleterious variant [16,17]. The high incidence of GC in Northern Portugal [18,19], the privileged industrial and economic situation of this region, that allowed inhabitants to settle and succeed, further supports a possible clustering of CDH1 germline variant carriers predisposed to early-onset DGC and/or LBC in this region [14,20].”
2) Authors write that in recent years northern Portugal had a fast population rise. A founder effect should be seen if this wave of immigration came in large part from a very isolated community. Do they have evidence for this? In case add reference.
Response 2: We thank Reviewer 2 for this question. We have partially addressed this point when replying to the previous comment. The answer to this particular point is rather complex. Indeed, the fast rise of the populations from the Porto district during the XX century is well documented by the official census of Portuguese population (“Instituto National de Estatística”). However, data for earlier periods, before the XIX, are not easily accessible. The historical information collected from several local organizations and information from the catholic church, point to a scattering of populations in small villages with the numbers of inhabitants fluctuating between one to four hundred.
We hope the reviewer understands these limitations and the impossibility of reporting better supported historical information due to data protection rules.
3) The authors write (page 2 line 97) of high GC incidence in Northern Portugal. Do they have supporting data for this? In case please cite. Finally, the conclusion mentions an unspecified clinical finding. What is that?
Response 3: We thank Reviewer 2 for the observation, and we addressed it already in the reply to comment 1 (please see above). To support the high incidence of gastric cancer in northern Portugal, and particularly in the Porto district, we added two references to the manuscript:
Added references:
Castro, C., Antunes, L., Lunet, N., et al. 25, 472-480 (2016).Cancer incidence predictions in the North of Portugal: keeping population-based cancer registration up to date. European Journal of Cancer Prevention PMID: 26317384 DOI: 10.1097/CEJ.0000000000000199
Castro C., Peleteiro B, et al. "Trends in Gastric and Esophageal Cancer Incidence in Northern Portugal (1994-2009) by Subsite and Histology, and Predictions for 2015". https://doi.org/10.5301/tj.5000542. First Published August 23, 2016 PMID: 27647232 DOI: 10.5301/tj.5000542
4) Question to Figure 1. Timeline of studies on c.1901C>T mutation. What is the advancement between 2003 and 2004? Are the families they are referring to, different or is the same family with a different classification?
Response 4: The publication of 2003, reports an isolated case of DGC at 32 years old, lacking family history of HDGC or other cancer-associated syndrome, carrying the CDH1 c.1901C>T variant. In 2004, a new publication reported a family fulfilling HDGC clinical criteria and a heavy family history of DGC carrying the exact same variant. These were the first two apparently unrelated individuals/families identified to carry the c.1901C>T variant. This has now been clarified in Figure 1.
We also took the opportunity to perform a few other amendments to improve the message of Figure 1 (Page 5).
5) Question to Page 5 line 217. please correct maker with marker
Response 5: We thank the Reviewer 2 for the observation, the correction was done on Page 8 line 250 of the new version of the manuscript (please see Rev 2- Comment 5).
6) Page 6 second paragraph (lines 238-266). Authors mention the SNP rs1801552 which is co-segregating with the mutation. Have the authors verified if the T allele of this SNP is altering splicing modulation sequences such as ESE, ESS, with available web algorithm?
Response 6: We thank the Reviewer 2 observation. The rs1801552 is a very frequent SNP in the all world populations and has been classified as a benign variant according to Clinvar and was revised by an FDA expert panel (https://www.ncbi.nlm.nih.gov/clinvar/variation/142770/). From a molecular standpoint, this variant has never been reported to impact protein function, mRNA expression or splicing and its frequency is equivalent in diseased and non-diseased populations, which is consistent with its benign classification. From our colony PCR sequencing, the vicinity of the rs1801552 SNP remained wild-type in all transcripts sequence, independently on whether the transcript was wild-type or mutant for the c.1901 position, as depicted in Supplemental Figure 3. Furthermore, we performed the same in silico analysis for the rs1801552 SNP, as used for the c.1901C>T alteration, using Netgene2 Server (http://www.cbs.dtu.dk/services/NetGene2/) (PMID: 2067018 DOI: 10.1016/0022-2836(91)90380-o) and we did not see a difference in the splicing prediction of both rs1801552 alleles.
7) While PTC bearing mutant allele would be subject to NMD still they do see some mutant RNA production and without using NMD-rescuing reagents (i.e. puromycin or PAXgene tubes). So the effect is partial. Further, authors should detail the ratio of down-regulation of the mutant RNA (approximately down to 30% when compared to wt in figure 2C).
Response 7: We thank and agree with Reviewer 2. Therefore, we considered this data, when describing the Allele-specific imbalance analysis results from Figure 2 Painel C (Page 10) and in the results section.
Page 9, Rev 2 - Comment 7: “This transcript abundance analysis supported an active degradation of the PTC bearing allele, leading to its reduction in approximately 66%, consistent with the truncating nature of the CDH1 c. 1901C>T variant.”
Alteration in Figure 2:
8) page 8 line 287. As above please define northern Portugal (eventually add map).
Response 8: As asked in previous comments, we added in Figure 1 (Page 5), a panel with a map of Portugal depicting the Porto district, known to present higher incidence of GC than the rest of the country, and also the region where the nine apparently unrelated families carrying the c.1901C>T reside.
Alteration in Figure 1:
DISCUSSION SECTION
9) In the discussion authors could expand the discussion on NMD effects possibly explaining why it appears to be partial in the data they show (see ref PMID 33928629 and PMID 33313762).
Response 9: Although a PTC is generated in the transcript derived from the c.1901T allele, the mRNA expression of this allele is 66% lower than the expression of the c.1901C wild-type allele (Figure 2 Painel X). Assuming the high efficiency of NMD in degrading PTC-bearing alleles, this may not seem an exceptional downregulation. However, we have seen similar results when using RNA extracted either from peripheral blood lymphocytes (PBLs) or normal gastric mucosa from patients carrying germline nonsense variants in CDH1 from different families, in two of our previous papers (PMID: 19965908; PMID: 18427545). The reason underlying the lack of complete downregulation of the PTC-bearing allele may result from a decalage between transcription and degradation. As none of these variants results in transcription inactivation, the expression of the mutant allele is likely similar to that of the wild-type. After transcription, splicing and the pioneer round of translation initiate, as well as degradation by NMD. This delay likely explains the residual detection of c.1901T variant-bearing transcripts.
To clarify this issue, we have added the following paragraph to the discussion:
Page 16, Rev 2- Comment 9: “We have published similar levels of downregulation of PTC-carrying alleles, when using RNA extracted either from peripheral blood lymphocytes (PBLs) or normal gastric mucosa, from patients carrying germline nonsense variants in CDH1 from different families, in two of our previous papers [28,30]. The reason underlying the lack of complete downregulation of the PTC-bearing allele by NMD, may result from a delay in degradation after transcription. Indeed, as none of the nonsense variants studied results in transcription inactivation, the transcription level of the mutant allele is likely similar to that of the wild-type. After transcription, splicing and the pioneer round of translation initiate, as well as degradation by NMD of PTC-carrying alleles [31]. This delay likely explains the residual detection of c.1901T variant-bearing transcripts in PBL samples from the c.1901C>T variant carriers. Regarding the impact that complete or incomplete downregulation of the mutant allele may have for disease phenotype and severity, very little is known, however it has been documented that certain genetic modifiers, and inter-individual variability in NMD efficiency between patients carrying identical mutations, may lead to differences in the disease severity and clinical phenotype [32].”
10) In the second paragraph of page 12 (line 396-401) authors may expand the discussion on the classification of missense variants affecting splicing (DOI 10.3390/genes9040216, PMID 30361419, PMID: 34270938)
Response 10: We understand the point of Reviewer 2 and we added a paragraph in the discussion alluding to the need of a wider characterization of the effects of apparent missense variants, and we cited the suggested references.
Pages 16 and 17, Revisor 2 - Comment 10: “Several publications have now demonstrated that a considerable fraction of variants, initially assigned as missense, are indeed splice-site variants. An example has been published for NF1, a gene causing Neurofibromatosis type 1 [34]. After applying in silico predictive tools to 41 variants previously classified as missense variants, authors demonstrated that over 45% of them were shown to affect splicing, and claim that such type of variants is frequently underscored if not analyzed in depth. These observations were further supported by studies using next generation sequencing technology [32] and high-throughput splicing reporter assays [35] to unravel the effect of single nucleotide variants, including missense, on splicing in other genes such as the von Willebrand factor gene and POU1F1, respectively. Altogether, there’s a need for improvement of in silico tools to predict cryptic splicing, and for the development of guidelines for formal analysis of splicing defects induced by SNVs, particularly synonymous and missense variants. These will certainly improve the current knowledge of the molecular mechanisms causing disease and will facilitate variant interpretation.”
11) Page 12 line 414. In here I'm not sure that the 6 bp differences in D16S3095 results correspond to three different mutational steps. Misalignment and slippage errors could also consist of two dinucleotide repeats or three dinucleotide repeats, therefore the mutation steps are not necessarily three. If authors have references on this topic please add.
Response 11: We understand and thank Reviewer 2 for this comment, and we provide a clarification below.
Although large mutational jumps might occur in short tandem repeats, the most common mutational process is through the gain or loss of one repeat fitting the postulated stepwise mutation model (PMID: 8401493; PMID: 8349120; PMID: 8159720).
In the specific case of family F7, and as indicated in lines 517-519, we assumed a recombination event as the source of haplotype divergence given that D16S3095, as well as the remaining downstream loci (D16S3106, D16S3018, D16S3049, D16S3098), all differ from the haplotypes identified in the other studied families. This assumption is therefore independent of the mutational model best suited to the D16S3095 polymorphic marker.
Still, as requested by the reviewer, we included in the text (lines 519-523) a reference (PMID: 8349120) for the assumed dinucleotide repeat mutational model.
Page 17, Rev 2 - Comment 11: “This can be demonstrated, firstly, by a 6 bp difference in D16S3095 (154 bp in F7 vs 148 bp in F1 and F2) corresponding to three mutational steps (three dinucleotide repeats) [39] and, secondly, because downstream markers deviate also from the common ancestral haplotype.”
12) Page 12 line 433: From where the exact year (1528) comes?
Response 12: We thank Reviewer 2 for the observation. The provided DHSMAP results (489,64 years) were obtained in 2018. Thus, subtracting 490 to 2018, we concluded that the TMRCA is from 1528. To make it clearer for the readers we detailed it in the text.
Page 17, Rev 2 - Comment 12: “This study estimates that the c.1901C>T variant has been segregating within the Portuguese population since the 16th century, and close to the year of 1528. This date was calculated by subtracting the estimated CDH1 allele age (490 years) to the year of 2018, when this analysis was performed (1528 = 2018 - 490 years), and assuming a 25 years generation time. These data may also indicate that more families are likely to be at risk and remain unidentified, to date.”
13) Although the mutation appears to be associated to a clinically aggressive phenotype it also shows some incomplete penetrance (overall 17 DGC+LBC in 58 carriers). The authors acknowledge this in the last discussion paragraph. Do they have data on the median age of healthy carriers that might disclose some hints on a combination of lower penetrance and age dependent penetrance (in case even comparing with other pathogenic CDH1 variants)?
Response 13: We thank Reviewer 2 for this observation and we altered the manuscript to address this subject. We compared the average age of and age-range (disease onset, genetic test, or last surveillance) for four groups in our cohort: 1) carriers presenting either DGC and/or LBC; 2) carriers with subclinical disease (foci in RRG or RRM); 3) carriers without foci detected in RRG and RRM specimens; 4) Asymptomatic carriers not submitted to risk reduction surgery.
Patients with either clinical or subclinical expression of disease presented on average age below 40 years old. A non-neglectable fraction of carriers (34,5%) rejected prophylactic measures and remain healthy to date with an average age of 42 years old, being the oldest non-affected carrier 83 years old. This supports the idea that c.1901C>T is a low penetrant variant, possibly regulated by other genetic modifiers that can either protect or promote disease development.
We added in the text the following paragraph:
Page 15, Rev 2 - Comment 13: “We further analyzed the average age and age-range of carriers upon disease onset, genetic test, or last surveillance, and separated the cohort (n=58 carriers) in four groups: 1) carriers presenting clinical expression of DGC and/or LBC; 2) carriers with subclinical disease (foci found in RRG or RRM); 3) carriers without foci detected in RRG and RRM specimens; 4) Asymptomatic carriers not submitted to risk reduction surgery. The average age of onset of the first group including all carriers with DGC and/or LBC (n=16) was 39 ± 13.98 years old [age range: 18-61 years old]. The average age at which subclinical disease was identified in the second group of asymptomatic carriers (n=18), upon risk-reduction surgery, was 36 ± 14 years old [age range: 14-63]. In the third group, carriers without subclinical disease (n=4), the average age at risk reduction surgery was 40 ± 14.9 years old [age range: 23-59]. The remaining asymptomatic carriers decided to avoid RRG and/or RRM (n=20). In these, follow-up was carried out either for DGC in both genders and for LBC in females, starting at an average age of 44 ± 22.7 years old [age range: 18-83] and to date none has developed clinical disease. These data support a high variability in age of disease onset and the incomplete penetrance in these families. The c.1901C>T is therefore a deleterious but low penetrant variant probably, due to the co-segregation with either protective or disease-predisposing genetic modifiers.”
n average age of 44 ± 22.7 years old [age range: 18-83] and to date none has developed clinical disease. These data support a high variability in age of disease onset and the incomplete penetrance in these families. The c.1901C>T is therefore a deleterious but low penetrant variant probably, due to the co-segregation with either protective or disease-predisposing genetic modifiers.”
